# Molecular analysis of acute pyelonephritis—excessive innate and attenuated adaptive immunity

Ines Ambite[1,*], Sing Ming Chao[2,*], Therese Rosenblad[1,3,*], Richard Hopkins[4], Petter Storm[1], Yong Hong Ng[2], Indra Ganesan[2], Magnus Lindén[5], Farhan Haq[1], Thi Hien Tran[1], Shahram Ahmadi[1], Bernett Lee[6], Swaine L Chen[7,8], Gabriela Godaly[1], Per Brandström[9,10], John E Connolly[4], Catharina Svanborg[1]

**This study investigated the molecular basis of disease severity in acute pyelonephritis (APN), a common and potentially life-threatening bacterial infection. Two cohorts of infants with febrile urinary tract infection were included. Renal involvement was defined by DMSA scans and molecular disease determinants by gene expression analysis and proteomic screens, at diagnosis and after 6 mo. Innate immune hyper-activation, systemically and locally in the urinary tract, was defined as a cytokine storm. Neutrophil degranulation and renal toxicity genes were strongly regulated, with overexpression in the APN group (first DMSA+). Adaptive immune attenuation in the APN group further supported the notion of an immune imbalance. DNA exome genotyping identified APN and febrile urinary tract infection as genetically distinct and scarring associated genes, but the activation of renal toxicity genes during acute infection was unrelated to the development of renal scarring. The results define APN as a hyper-inflammatory disorder with the characteristics of a cytokine storm combined with adaptive immune attenuation. The findings are consistent with innate immune dysfunctions and neutrophil disorders identified as determinants of APN susceptibility in genetic models.**

## Introduction

The susceptibility to infection is ultimately controlled by the efficiency of the antimicrobial host defense in infected individuals and at the population level. Rare, primary immune-deficiencies increase the individual risk of selected viral, bacterial, or parasitic infections (Fodil et al, 2016) and a lack of herd immunity explains failures to combat emergent infections (Tangye et al, 2020). The susceptibility to common bacterial infections such as pneumonia, diarrhea, or urinary tract infection (UTI) varies extensively in the general population (Zupan, 2005; Telenti & di Iulio, 2020), and differences in immune control are expected to occur (Frendeus et al, 2000; Fischer et al, 2010; Ragnarsdottir et al, 2010), but the molecular basis of susceptibility and their underlying gene targets have not been identified. The existence of shared molecular susceptibility determinants is even doubted because of the prevalence and complexity of these infections.

UTIs affect more than 150 million adults each year and 5% of children below the age of 12 yr (Shaikh et al, 2008; Flores-Mireles et al, 2015). Acute pyelonephritis (APN) is characterized by high fever, general malaise, and loin pain, and urosepsis is a major cause of mortality worldwide (Wagenlehner et al, 2015; Mattoo et al, 2021). Up to 30% of children with APN develop renal scarring (Hoberman et al, 2003), associated with long-term morbidity (Geback et al, 2015; Swerkersson et al, 2017). Yet, because of poor resolution of molecular disease determinants that define susceptibility and long-term outcomes, the management of APN and assessment of clinical risk are often unsatisfactory. Genetic screens have identified specific molecular disease determinants of APN severity, and polymorphic genes in patients with APN include *TLR4* and the chemokine receptor *CXCR1* (Liang et al, 2019), but the molecular basis of APN and renal scarring has not been comprehensively investigated.

This study enrolled two cohorts of infants with a first febrile UTI episode. Streamlined clinical protocols were used to select the patients and global gene expression analysis and proteomic screening, was used at enrolment and follow-up after 6 mo. The results identify APN as a disease characterized by excessive innate immune activation with a cytokine storm profile, neutrophil dysfunctions, and inhibition of adaptive immunity, in a genetically distinct subset of patients.

[1]Division of Microbiology, Immunology and Glycobiology, Department of Laboratory Medicine, Lund University, Lund, Sweden  [2]Duke-National University of Singapore Academic Clinical Program, Pediatric Nephrology Service, KK Women's and Children's Hospital, Singapore, Singapore  [3]Department of Pediatrics, Lund Children's Hospital, Lund, Sweden  [4]Institute of Molecular and Cell Biology, Agency for Science, Technology and Research, Singapore, Singapore  [5]Department of Pediatrics, Halland Hospital, Halmstad, Sweden  [6]Singapore Immunology Network, Agency for Science, Technology and Research, Singapore, Singapore  [7]Laboratory of Bacterial Genomics, Genome Institute of Singapore, Agency for Science, Technology and Research, Singapore, Singapore  [8]Infectious Diseases Translational Research Program, Department of Medicine, National University of Singapore, Singapore, Singapore  [9]Pediatric Uro-Nephrology Center, Queen Silvia's Children's Hospital, Gothenburg, Sweden  [10]University of Gothenburg, Gothenburg, Sweden

Correspondence: catharina.svanborg@med.lu.se
*I Ambite, SM Chao, and T Rosenblad are joint first authors

# Results

Two prospective, analytical cohort studies were conducted to identify molecular susceptibility determinants and disease mechanisms in febrile UTI (Figs 1 and S1). Infants and young children with unexplained high fever and suspected febrile UTI were enrolled in Singapore and Sweden, respectively, to include populations with different genetic backgrounds (Figs S2 and S3). Enrolment was restricted to the first episode of febrile UTI in each patient, to capture the primary response to infection (Tables S1 and S2). A diagnosis of febrile UTI was confirmed by a positive urine culture (single growth > $10^4$ cfu/ml in catheterized urine or > $10^5$ cfu/ml in mid-stream clean-catch urine), a temperature >38.5°C, and pyuria and supported by increased C-reactive protein (CRP) levels (Xu et al, 2014). The gender distribution was similar to that in other studies of uncomplicated febrile UTI (Shaikh et al, 2008) (Figs 1C and S1). Most infections were caused by *Escherichia coli* (Figs 1D and S3). Bacterial genome sequencing identified a highly conserved, virulent genotype, including the pap operon, encoding P fimbriae (74%), the fim gene cluster (95%), and hemolysin (30%) (Fig S4). There was no association of bacterial genotype to first DMSA or second DMSA outcome.

A diagnosis of APN was assigned to 61/111 children in Cohort I, and 36/52 children in Cohort II based on a positive first DMSA scan within 5–7 d of enrolment (Figs 1A and S1). Renal scarring was defined by a positive second DMSA scan, in 23/52 patients in Cohort I, and 5/29 patients in Cohort II. 10 patients from Cohort I had functionally important and/or bilateral scars associated with reduced creatinine clearance (Yiee et al, 2010). Vesicoureteric reflux (VUR) was detected in 17.7% of 96 patients investigated by micturating cystourethrogram (MCU) including 12 with high-grade reflux (grades III or IV) (Fig 1B). Renal scarring occurred in eight patients with high-grade reflux (36.4%), four with low-grade reflux, and 10 without reflux (Liang et al, 2019). There were seven cases of recurrent UTI, three patients had a recurrent UTI after 6 mo, and four had a recurrent UTI within 2 mo of the first UTI episode, where the second DMSA was performed 4 mo after second UTI episode.

## Disease severity defined by gene expression analysis—differences between febrile UTI and APN

To characterize the systemic disease response, peripheral blood RNA samples were subjected to genome-wide transcriptomic analysis, comparing gene expression profiles at enrolment to the 6-mo follow-up. The APN group (first DMSA+) was further compared with the group with febrile UTI (first DMSA−) (Figs 2 and S5). Gene expression was strongly activated at enrolment in Cohorts I and II, as illustrated by volcano plots in Fig 2A. Principal component analysis clearly separated gene expression profiles at enrolment from the follow-up samples in both cohorts, defining the acute response as disease related (Figs 2B and S6).

The top scoring canonical pathways overlapped extensively between the cohorts, including the neutrophil degranulation pathway ($P = 10^{-61}$), the pathogen-induced cytokine storm pathway ($P = 10^{-14}$), and the neuro-inflammation pathway ($P = 10^{-9}$) (Figs 2C and S5). The cytokine storm profile of activated genes, conserved in Cohorts I and II, suggested a shared molecular basis of disease (Fig 2D). The neutrophil

surface lipoprotein *CD177* (fold change [FC] = 45.1), first described in neonatal neutropenia (Lalezari et al, 1971) and the mast cell protein *MCEMP1* (FC = 11.4), involved in mast cell maturation, were identified as the most strongly differential expressed genes in both cohorts (Table S3). CD177, or NB1, is a neutrophil surface adhesion molecule, which interacts with the beta-2 integrin heterodimer (CD11b/CD18 or Mac-1 or complement receptor 3) and mediates neutrophil activation, degranulation, and superoxide production (Jerke et al, 2011).

In addition, gene expression was more strongly differentially expressed in the APN (first DMSA+) than in the febrile UTI group (first DMSA−) ($P < 0.001$, $\chi^2$ test) as illustrated by heatmaps (Figs S5 and S7) and volcano plot (Fig 2E and Table S4). Canonical pathway analysis (Fig 2F) identified major innate immune signaling pathways as more strongly activated in APN (first DMSA+) than febrile UTI (first DMSA−), including inflammatory cell recruitment and neutrophil degranulation, TLR signaling, interleukin signaling (IL-6, TNF), IL-1, and inflammasome family signaling and complement regulation pathways (Figs 2E and S8).

The APN-specific gene expression profile further identified adaptive immunity genes as attenuated in both cohorts (Fig S5). Unexpectedly, the NF-κB pathway was inhibited in the first DMSA+ group, as were adaptive immune functions including TCR signaling and IL-2 expression. The T-cell exhaustion pathway was activated and the CTLA4 inhibitor of cytotoxic T-cell function ($P = 10^{-54}$) (Figs 2C and S5). B-, T-, and NK-cell development genes were broadly inhibited (*CD3, lambda 5, ZAP70, ICOS*, and the B-cell linker *BLNK*), as were genes involved in primary immunodeficiency signaling. $Ca^{2+}$-dependent T-lymphocyte apoptosis was suppressed, as were Fc receptors and immunoglobulin genes (Table S5), possibly impairing the communication between innate and adaptive immunity.

The results identify a shared, hyper-activated innate immune response in patients with febrile UTI and a specific over-activation profile in the APN group, with the characteristics of a cytokine storm. There was also evidence of an adaptive immune attenuation disorder in the first DMSA+ group. This gene expression profile was detected in both study cohorts, supporting the presence of an innate and adaptive imbalance in patients with febrile UTI and especially acute renal involvement.

## Evidence of a local cytokine storm in the urinary tract

The urinary tract response to infection was further analyzed in Cohort I, by proteomic analysis of 84 cytokines, cytokine receptors and molecules associated with kidney injury. Urine samples obtained during acute disease were compared with samples obtained at follow-up (Fig 3A). The proteomic screen detected increased concentrations of 43 proteins, reflecting the extensive local immune response to infection (Fig 3B).

The cytokine response was functionally characterized as a cytokine storm (Fig 3C), consistent with the gene expression analysis in Fig 2D (Fajgenbaum & June, 2020). Hyper-activated chemokines included CXCL-8, involved in neutrophil recruitment and activation, regulators of innate immunity such as IL-6, IL-1, and downstream pathways controlled by IL-1β, IL-6, MCP-1, and IP-10, which is IFN-γ induced (Fig 3B). The results suggest that an excessive innate immune response, characterized as a cytokine storm occurs locally, in the urinary tract.

**A** Enrollment and Outcomes

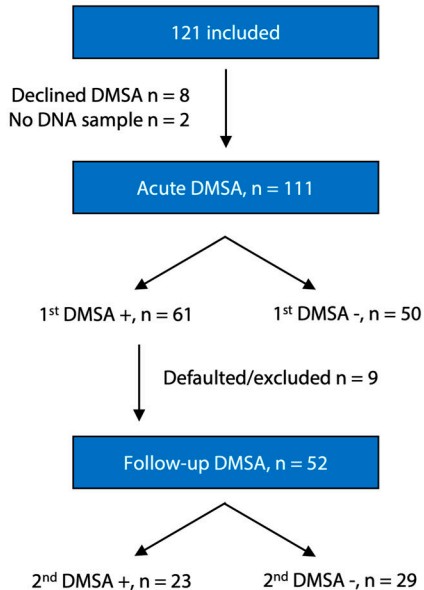

**B** Examination and Imaging Performed

| | No. of Patients (%) |
|---|---|
| 1st DMSA scan (n = 111) | |
| Positive: Acute pyelonephritis | 61 (55) |
| Negative: No acute pyelonephritis | 50 (45) |
| Renal ultrasound (n = 111) | |
| Normal | 83 (75) |
| Abnormal: | 28 (25) |
| Pelvicalyceal dilatation, ureteric dilatation, thickened uroepithelium, increased echogenicity, swollen kidneys | |
| Micturating cystourethrogram (n = 96) | |
| No VUR | 79 (82) |
| VUR[a] | 17 (18) |
| Grade I-II | 5 (29) |
| Grade III-IV | 12 (71) |
| Grade V | 0 |
| 2nd DMSA scan (n = 52) | |
| Negative: No scars | 29 (56) |
| Positive: Scars | 23 (44) |
| Minor | 13 |
| Bilateral | 2 |
| Unilateral, differential function <42% | 8 |

[a]significant association with 1st DMSA+, P=0.002

**C** Demographic Characteristics

**D** Urine Cultures

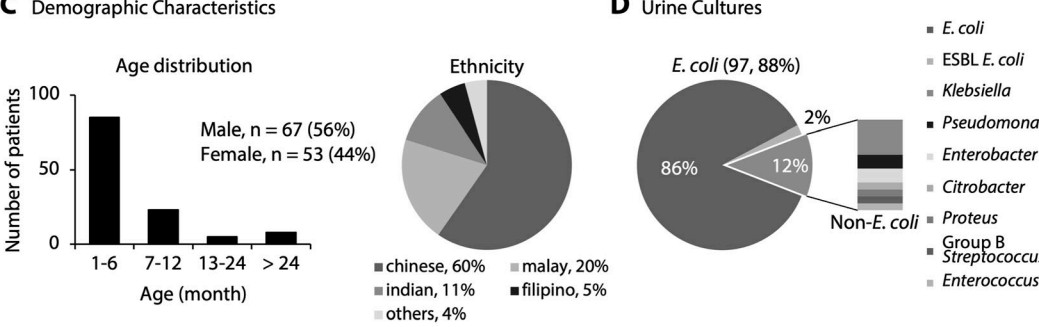

**E** C-reactive Protein Levels

**F** DMSA Results, Acute and Follow-up

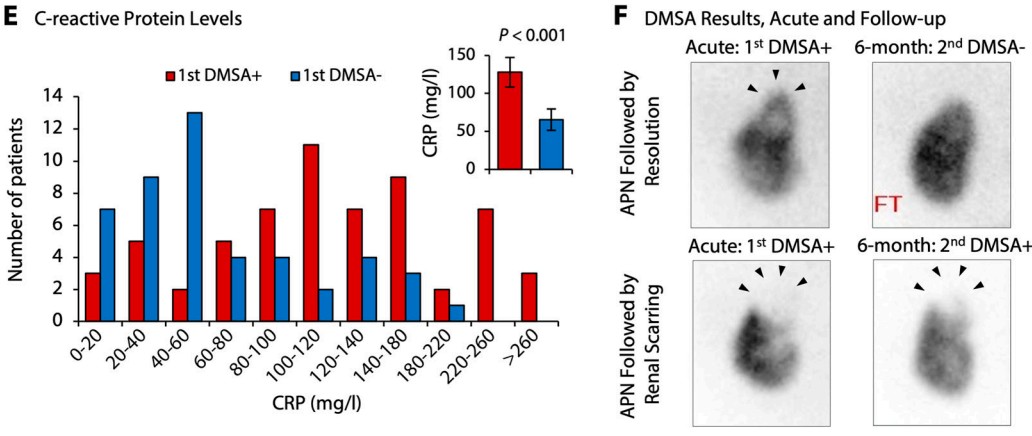

**Figure 1. Clinical study variables in children with a first febrile UTI episode.**
**(A)** Scheme of patient enrolment and clinical outcomes in Cohort I. APN was defined by a positive first DMSA scan. **(B)** Outcome of investigations using imaging techniques. Renal ultrasound or micturating cystourethrogram did not distinguish the patients with APN or renal scarring from the remaining group with febrile UTI. VUR, vesicoureteric reflux. **(C)** Demographic data of Cohort I (see also Table S1). **(D)** Urine culture results, identifying *E. coli* as the causative organism in most of the patients. One patient with first DMSA+ had a negative culture. **(E)** Elevated C-reactive protein levels in the first DMSA+ group compared with the first DMSA− group (P < 0.001, Mann–Whitney test). Insert denotes means and 95% confidence intervals. **(F)** Example of acute renal involvement (top panels), defined by a positive first DMSA scan, followed by resolution after 6 mo. And evidence of renal scarring (lower panels), defined by positive first DMSA scan at enrolment and a positive second DMSA scan after 6 mo. Arrows indicate area of renal defect. See also corresponding data for Cohort II (Fig S1 and Table S2).

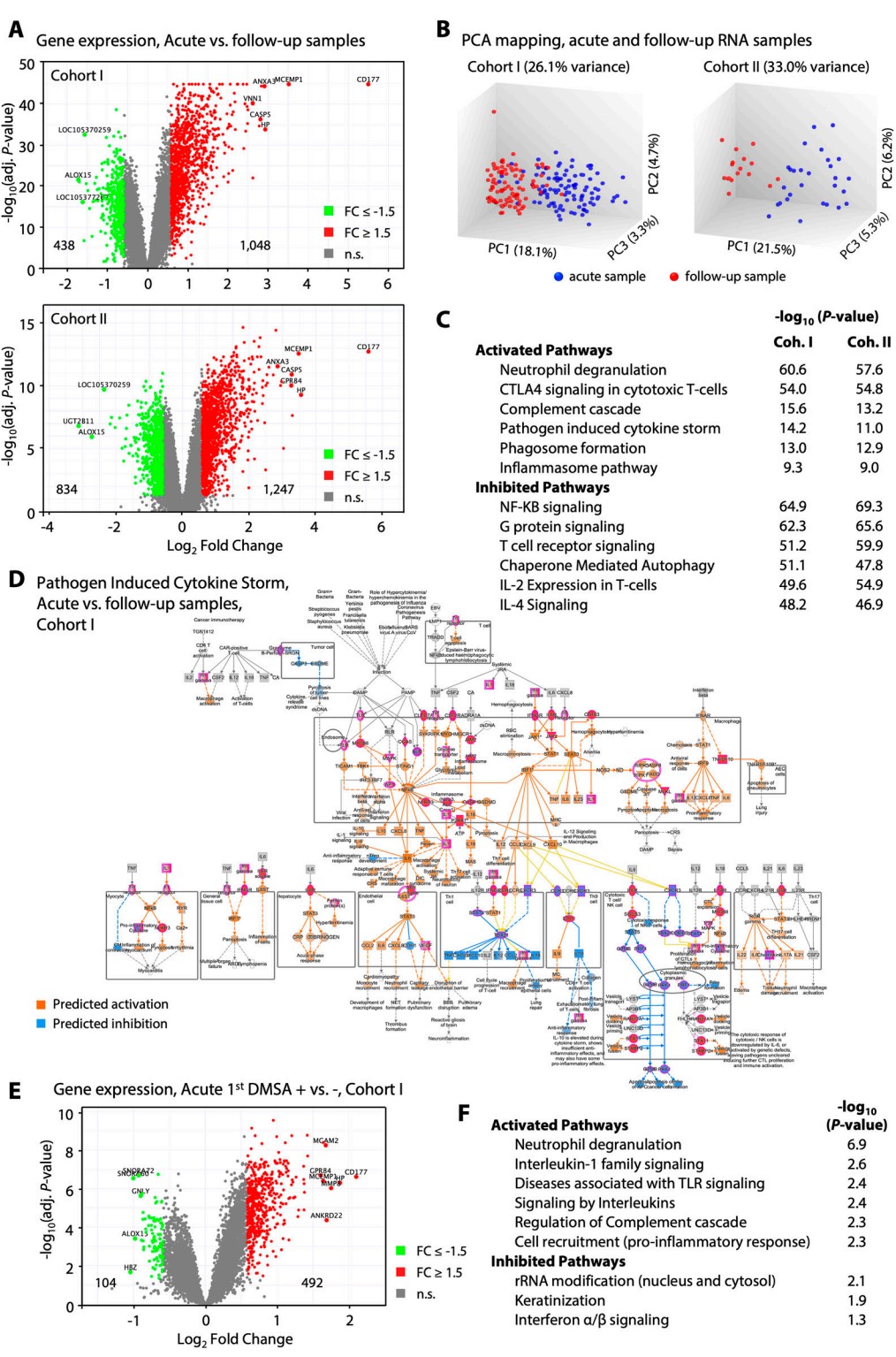

**A** Gene expression, Acute vs. follow-up samples

Cohort I

FC ≤ -1.5 (green), FC ≥ 1.5 (red), n.s. (grey)

438 | 1,048

Cohort II

834 | 1,247

**B** PCA mapping, acute and follow-up RNA samples

Cohort I (26.1% variance) — PC1 (18.1%), PC2 (4.7%), PC3 (3.3%)

Cohort II (33.0% variance) — PC1 (21.5%), PC2 (6.2%), PC3 (5.3%)

● acute sample  ● follow-up sample

**C**

|  | -log$_{10}$ ($P$-value) | |
| --- | --- | --- |
|  | **Coh. I** | **Coh. II** |
| **Activated Pathways** | | |
| Neutrophil degranulation | 60.6 | 57.6 |
| CTLA4 signaling in cytotoxic T-cells | 54.0 | 54.8 |
| Complement cascade | 15.6 | 13.2 |
| Pathogen induced cytokine storm | 14.2 | 11.0 |
| Phagosome formation | 13.0 | 12.9 |
| Inflammasome pathway | 9.3 | 9.0 |
| **Inhibited Pathways** | | |
| NF-κB signaling | 64.9 | 69.3 |
| G protein signaling | 62.3 | 65.6 |
| T cell receptor signaling | 51.2 | 59.9 |
| Chaperone Mediated Autophagy | 51.1 | 47.8 |
| IL-2 Expression in T-cells | 49.6 | 54.9 |
| IL-4 Signaling | 48.2 | 46.9 |

**D** Pathogen Induced Cytokine Storm, Acute vs. follow-up samples, Cohort I

■ Predicted activation
■ Predicted inhibition

**E** Gene expression, Acute 1st DMSA + vs. -, Cohort I

FC ≤ -1.5 (green), FC ≥ 1.5 (red), n.s. (grey)

104 | 492

**F**

|  | -log$_{10}$ ($P$-value) |
| --- | --- |
| **Activated Pathways** | |
| Neutrophil degranulation | 6.9 |
| Interleukin-1 family signaling | 2.6 |
| Diseases associated with TLR signaling | 2.4 |
| Signaling by Interleukins | 2.4 |
| Regulation of Complement cascade | 2.3 |
| Cell recruitment (pro-inflammatory response) | 2.3 |
| **Inhibited Pathways** | |
| rRNA modification (nucleus and cytosol) | 2.1 |
| Keratinization | 1.9 |
| Interferon α/β signaling | 1.3 |

**Figure 2. Gene expression analysis of RNA samples from Cohorts I and II.**
In this longitudinal study of febrile UTI, RNA was extracted at enrolment, and at follow-up after 6 mo. **(A)** Visualization of the acute response to infection in the two study cohorts. Strong up-regulation of gene expression was apparent in all patients with febrile UTI, in both cohorts. Volcano plots showing 1,048 up-regulated (71%) versus 438 down-regulated genes in Cohort I and 1,247 up-regulated (60%) versus 834 down-regulated genes in Cohort II (cutoff FC ≥ 1.5, adj. $P$ < 0.05 compared with follow-up; red, up-regulation; green, down-regulation; grey, non-significant). **(B)** Principal Component Analysis plot of RNA samples. Principal Component Analysis mapping clearly separated the acute from the follow-up samples in each Cohort (PC1 18% and 22%, respectively). **(C)** Functional analysis of genes regulated during acute febrile UTI compared with follow-up samples. Pathway analysis revealed strong activation of innate immune response genes, including neutrophil degranulation, complement cascade, pathogen-induced cytokine storm, phagocytosis, and inflammation. Inhibited pathways were involved in adaptive immune responses, including T-cell receptor and NF-κB signaling, and IL-2 and IL-4 response pathways. The top regulated canonical pathways were shared between Cohorts I and II at inclusion, suggesting a conserved disease response. $P$-values were generated for each cohort separately, for acute compared with follow-up samples. **(D)** Activation of the pathogen-induced cytokine storm signaling pathway during the acute febrile UTI response. Notably, Toll-like receptors, interleukins, and interleukin receptors, interferon gamma receptors were activated. **(E)** Evidence that gene expression is more strongly activated in the first DMSA+ group. Volcano plot showing 492 up-regulated (83%) versus 104 down-regulated genes, directly comparing the first DMSA+ and first DMSA− groups in Cohort I (cutoff FC ≥ 1.5, adj. $P$ < 0.05; red, up-regulation; green, down-regulation; grey, non-significant). **(E, F)** Functional categorization of the response identified in (E). Innate immune functions were activated in the first DMSA+ group compared with the DMSA− group, including neutrophil degranulation, IL-1, and interleukins signaling, TLR activation, and pro-inflammatory cells recruitment.

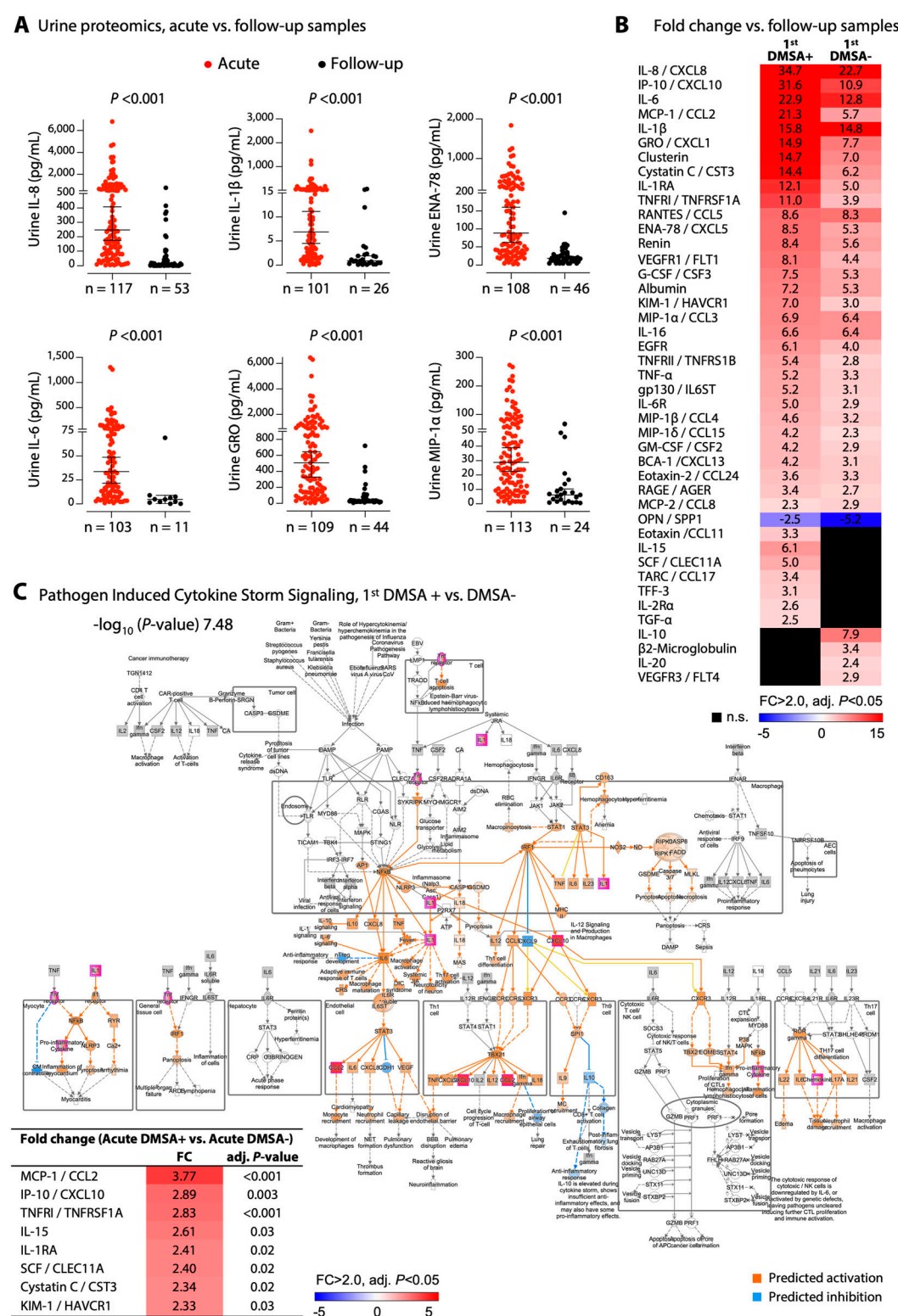

**Figure 3. Cytokine storm during APN.**
**(A)** Examples of hyper-activated cytokines in acute urine samples compared with the follow-up samples. The most strongly activated cytokines in Cohort I are shown (IL-8, IL-6, IL-1β, GRO ENA-78, and MIP-1α). Lines represent the medians with 95% confidence intervals. Two-way ANOVA with Šidák's multiple comparisons test of log$_{10}$-normalized concentrations values. **(B)** Urine levels of 85 protein markers were quantified and compared between acute and follow-up samples, FC of concentrations. Cohort I, acute urine samples (n = 118–25) and follow-up urine samples (n = 73–7), Mann–Whitney test. **(C)** Acute regulation of pathogen-induced cytokine storm proteins in urine, comparing acute first DMSA+ and DMSA− samples.

## Hyper-activation of neutrophil degranulation in febrile UTI and APN

Neutrophil degranulation was identified as the most strongly activated pathway in febrile UTI and APN ($P = 10^{-60}$) (Fig 2C and F). Neutrophil degranulation pathway genes (n = 138) were more strongly regulated in APN (first DMSA+) than in febrile UTI (first DMSA–) (Fig 4A and B). Based on the analysis of neutrophil function by Naranbhai et al, activated genes were shown to represent multiple stages of neutrophil development and activation (Naranbhai et al, 2015), including chemokine-induced migration, granule formation, and degranulation; phagocytosis; NETosis; and immune crosstalk (Fig 4C). *CD177* was the most strongly up-regulated gene in both cohorts (FC = 45 and 48, respectively, Table S3) The CD177 response was confirmed at the protein level, by quantifying CD177 concentrations in acute urine samples (paired samples, $P < 0.001$ compared with follow-up samples, Fig 4D). Receiver operating characteristics curves of expression values identified *CD177* as a highly efficient classifier of acute febrile UTI (area under the curve > 0.980, $P < 0.001$ in both cohorts, compared with the follow-up samples) (Figs 4E and S9).

## Neutrophil degranulation genes as predictors of acute renal involvement in children with febrile UTI

To address if the most strongly regulated genes in this patient group might serve as markers of acute disease severity, the Boruta feature selection algorithm (Kursa & Rudnicki, 2010) was used to interrogate the neutrophil genes (n = 565) as markers of APN. Patients in the APN group (first DMSA+) were compared with the febrile UTI group without renal involvement (first DMSA–), and 34 genes were shown to predict the first DMSA outcome, with 24 confirmed important attributes (Fig 4F). Using the Receiver operating characteristics curve analysis, *MAP2K2* expression was the most strongly predictive classifier of acute renal involvement (first DMSA+, AUC 0.806, Fig S9), followed by *PGM2*, *DSP*, *ABCA13*, *SVIP*, and *FLG2*. By combining a four-gene signature (*MAP2K2*, *PGM2*, *SVIP*, and *FLG2*), an AUC of 0.944 was achieved (95% CI 0.897 to 0.991, Fig 4G). MAP2K2 (MEK2) is involved in NETosis regulation, PGM2 (phosphoglucomutase 2) is an enzyme present in ficolin-rich granules, SVIP (small VCP-interacting protein) is a tertiary granule membrane protein, and FLG2 (filaggrin 2) is a tertiary granule lumen protein. In contrast, there was no evidence that the activation of neutrophil degranulation genes was predictive of renal scarring (second DMSA+) (Fig 4H). Instead, eight interferon-related genes were identified as associated with scarring by the Boruta feature selection method (Fig 4H).

## Neutrophil accumulation and CD177 staining in infected kidneys

A murine model of APN was used to investigate if CD177 and neutrophil degranulation are affected in infected renal tissues. Female *Irf3*$^{-/-}$ mice, which are highly susceptible to APN (Puthia et al, 2016), were intra-vesically infected with the uropathogenic *E. coli* strain CFT073 and euthanized after 24 h or 7 d (Fig S10). Neutrophil numbers in urine increased rapidly and remained elevated until day 7 and significant neutrophil staining was detected

along the renal pelvis (Fig S10). Bacterial counts in urine showed similar kinetics and tissue staining–detected bacteria in the same areas as the neutrophils, resulting in micro-abscess formation along the renal pelvis (Fig S10).

Gene expression analysis identified the neutrophil degranulation pathway as the top regulated pathway in infected kidneys, 7 d post infection, compared with uninfected controls (Fig S11). *Cd177* was strongly activated and immunohistochemistry staining of kidney tissue sections detected a significant accumulation of CD177 protein in infected kidneys, after 24 h and 7 d, compared with uninfected controls (Fig S11).

## Renal toxicity defined by gene expression analysis

APN is an important cause of acute renal tissue damage and renal scarring (Roberts, 1999), suggesting that infections affect the expression of renal damage-associated genes. In the acute patient samples, functional analysis identified 215 strongly activated renal damage-associated genes, compared with follow-up samples (Fig 5A). Top regulated genes were associated with neutrophil recruitment and accumulation, the accumulation and activation of lymphocytes, and NK-cell toxicity. These functions were more strongly regulated in the APN (first DMSA+) than in the febrile UTI (first DMSA–) group, consistent with the difference in renal involvement detected by the DMSA scans (Fig 5A).

Furthermore, the analysis of renal toxicity genes activated during acute infection did not identify a specific profile in the subset of patients who developed renal scarring, suggesting that the renal toxicity genes regulated in the APN group during acute infection, are not associated with renal scarring. Interestingly, there was no evidence of a lasting systemic hyper-inflammatory response in the renal scarring group and gene expression analysis identified one differentially regulated gene associated with renal scarring (Fig 5B). The local response, defined by cytokine levels in urine at the time of follow-up, showed minor changes. At inclusion, urine osteopontin levels were higher in samples from the scarring group than in those who resolved ($P = 0.05$, Fig S12).

## DNA-based association studies for APN and renal scarring

To identify genetic variants associated with APN or renal scarring, patient DNA was subjected to exome genotyping. Genes associated with acute DMSA outcomes were identified by computing odds risk ratios (ORRs) for individual genotypes, comparing the first DMSA+ with the first DMSA– group (Fig S13). Furthermore, allele frequencies were calculated and compared between the first DMSA+ and first DMSA– groups (Tables S6 and S7). The call rate across the entire cohort was >99.8% and the minor allele frequency 20.3%, leaving 35,008 SNPs with at least one alternative allele.

By calculating the ORRs for the entire data set, 714 genes that were identified as significantly associated with the first DMSA outcome in Cohort I (cutoff $P < 0.05$, range $\log_2$ORR 3.6–[–3.9], Fig S13). Eucledian ranking was able to correctly discriminate first DMSA positivity with 98.2% accuracy, based on first DMSA ($P < 0.01$) and the t-distribution Stochastic Neighbor Embedding (tSNE) algorithm was able to correctly distinguish positive first DMSA individuals with an accuracy of 99.1% (Fig S13). 824 genes were identified as significantly

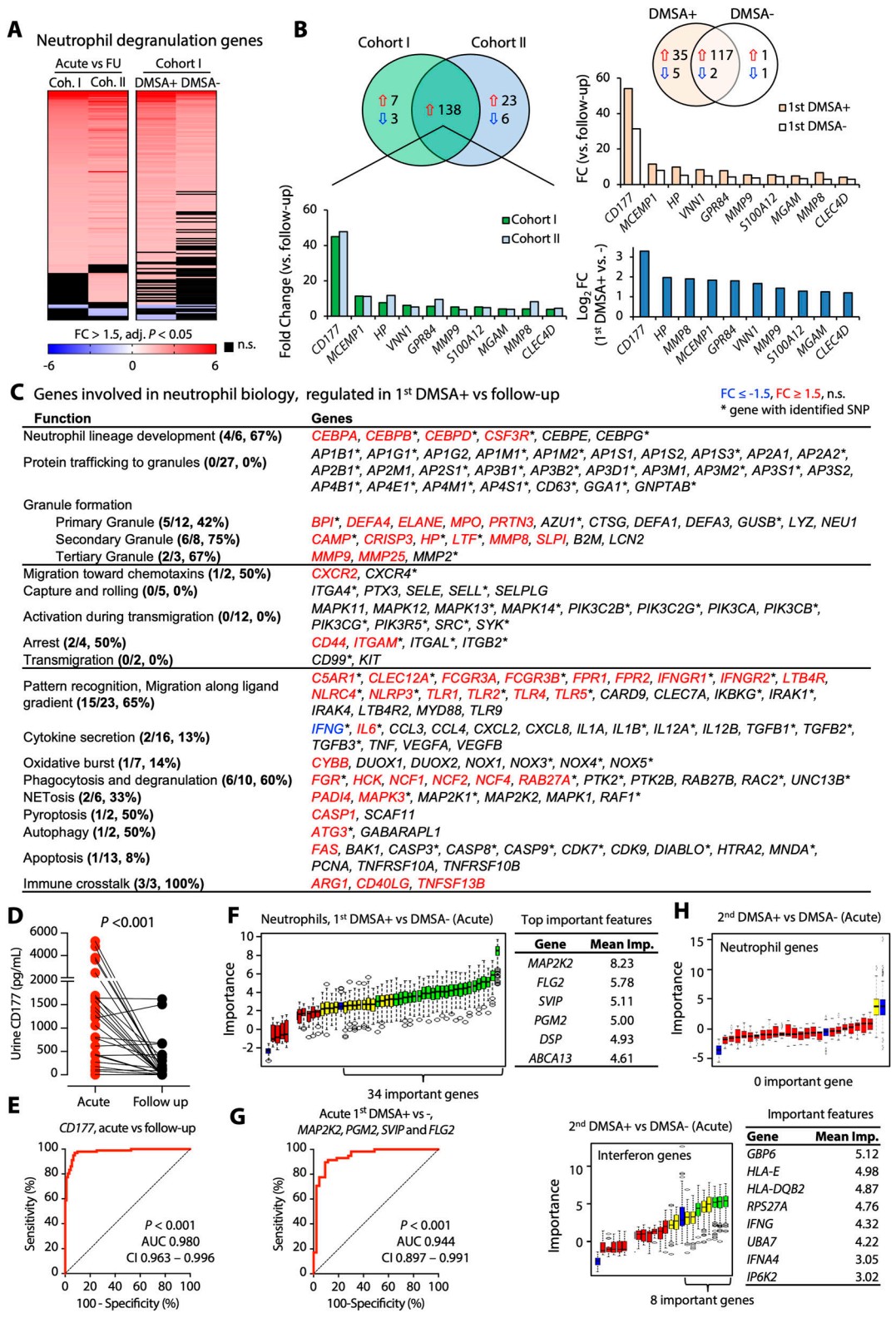

**Figure 4. Neutrophil-related genes predict febrile UTI and APN.**
**(A)** Visualization of the acute regulation of neutrophil degranulation genes in the two study cohorts (cutoff FC > 1.5, adj. *P* < 0.05 compared with follow-up; red, up-regulation; blue, down-regulation; black, non-significant). **(B)** Strong up-regulation of gene expression was apparent in all patients with febrile UTI, in both cohorts (n = 138 genes). In addition, a subset of genes was specifically regulated in the first DMSA+ group compared with the first DMSA− group (n = 37 DMSA+ specific genes). Top regulated genes included *CD177* (NB1) a specific marker of neutrophil adhesion and transmigration, *MCEMP1* a transmembrane protein expressed by mast cells, *HP* (haptoglobin)

associated with the first DMSA outcome in Cohort II (cutoff *P*-value <0.01, range log$_2$ORR 5.8–[–5.2]). Comparisons with the gnomAD database identified 473/714 genes in Cohort I and 386/824 genes in Cohort II, as significantly first DMSA associated (Fig S13).

A subset of 67 polymorphic genes associated with first DMSA outcome, was shared, in the two cohorts (Fig S14). A reconstructed functional network of polymorphic genes potentially affecting the host response to infection, included *CREB*, solute carriers, neutrophil migration, renal lesions and the LPS-responsive *BACH2* transcriptional regulator, potentially tying innate immunity to B-cell maturation and regulatory T$_{reg}$ cell function (Afzali et al, 2017).

A further analysis identified 622 genes as significantly associated with the second DMSA outcome (cutoff *P* < 0.05, range log$_2$ORR 4.4–[–4.4], Fig S15). Eucledian ranking was able to correctly discriminate second DMSA positivity with 98.2% accuracy and the tSNE algorithm correctly distinguished second DMSA positive individuals with an accuracy of 99.1%. MCU and US outcomes showed no association with the genetic profile (Fig S15). The results identify, for the first time, distinct genetic signatures in first DMSA+ compared with first DMSA– patients with febrile UTI, and in the subset of patients in the first DMSA+ group, who develop renal scarring, compared with those who resolve.

# Discussion

This study identified molecular disease determinants of APN, a common and severe bacterial kidney infection. Clinical samples were examined longitudinally in two cohorts of infants with their first febrile UTI episode, and the host response to infection was quantified by gene expression analysis and proteomic profiling. Genetic determinants of disease susceptibility were further investigated by exome sequencing, identifying a shared molecular basis of disease, with characteristics of a cytokine storm. These findings add APN to the group of severe infections that includes COVID-19 and bacterial sepsis, where excessive innate immune activation has been detected (de Jesus et al, 2015; Zanza et al, 2022). Gene expression data and proteomic analysis place the origin of the cytokine storm in the urinary tract and functional analysis further identified neutrophil degranulation as over-activated and neutrophil degranulation genes as highly polymorphic, consistent with dysfunctional neutrophil responses to infection (Frendeus et al, 2000; Fischer et al, 2010; Puthia et al, 2016). The excessive acute innate immune response was accompanied by adaptive immune attenuation and T-cell exhaustion, suggesting that an immune imbalance may be affecting both innate and adaptive immunity in this patient group.

The molecular basis of APN has been extensively studied in the murine UTI model and the importance of innate immunity for antibacterial defense is well established (Fischer et al, 2010; Gluba et al, 2010; Puthia et al, 2016; Ambite et al, 2021). In early studies, preceding the identification of Toll-like receptors and specific innate immune pathways, the neutrophil response to infection was shown to affect the efficiency of bacterial clearance and tissue pathology in infected kidneys (Poltorak et al, 1998). Excessive neutrophil infiltration in *Irf3*$^{-/-}$ mice is associated with severe acute disease and tissue pathology and neutrophil retention in the kidneys of the *Cxcr1*$^{-/-}$ mice is associated with renal scarring (Frendeus et al, 2000; Fischer et al, 2010; Puthia et al, 2016). Bacterial clearance is also impaired in *Irf3*$^{-/-}$ and *Cxcr1*$^{-/-}$ mice, suggesting that the recruited neutrophils might be functionally impaired and unable to carry out their protective effector functions in infected kidneys. In addition, data obtained in the murine UTI model confirmed that neutrophil degranulation is strongly affected locally in infected tissues of APN-prone mice, and CD177 staining was markedly increased, supporting the clinical observations made in this study.

Whereas neutrophil degranulation was identified as the most strongly activated pathway, genes regulating other immune cell types were also affected. The proteomic analysis identified the M1 macrophage polarizing and pro-inflammatory protein MCP-1/CCL2 (Kanda et al, 2006; Moore et al, 2015; Gschwandtner et al, 2019), as about fourfold higher in the APN (first DMSA+) compared with the febrile UTI (first DMSA–) group. In addition, the M1 macrophage-inducing protein osteopontin (OPN), which promotes fibrosis (Xu et al, 2023), was elevated in the second DMSA+ patients, suggesting that M1 polarizing macrophages residing in the kidney might be involved in the scarring process. Renal resident macrophages have been proposed to play an important role in renal scarring in C3H/HeOuJ mice (Li et al, 2017), cystitis (Lacerda Mariano et al, 2020), and undermining adaptive immunity (Mora-Bau et al, 2015).

The identified DNA variants provide further mechanistic insights into APN susceptibility, discriminating patients with acute renal involvement or renal scarring, from the remaining patients with febrile UTI. In previous studies, single gene deletions affecting innate immunity were shown to increase or reduce APN susceptibility, and polymorphisms affecting *CXCR1*, *TLR4*, *IRF3*, *CXCL8*, and *VDR* associated with APN susceptibility in clinical studies (Hagberg et al, 1984; Lundstedt et al, 2007; Liang et al, 2019). A three-generation family study supported the inheritance of APN susceptibility in both male and female family members of children with APN and the function of

secondary granule molecule released by activated neutrophils, *VNN1* (vanin 1) involved in hematopoietic cell trafficking, *GPR84* (neutrophil chemotaxis), *MMP9* (neutrophil activation and migration). Expression of those genes was higher in patients with APN than in patients with febrile UTI without renal involvement. **(C)** Details of neutrophil biology genes compiled by Naranbhai et al and their regulation in APN patients (Naranbhai et al, 2015). Red are up-regulated, and blue are down-regulated genes. Asterisk mark genes with identified SNPs in the genome sequencing data. **(D)** ELISA assay of CD177 responses in urine, in acute compared with follow-up samples in Cohort II. Lines represent paired samples (*P* < 0.001, Wilcoxon matched-pairs signed rank test). **(E)** Receiver operating characteristic curve of *CD177* expression values, in acute compared with follow-up RNA samples. The area under the curve was 0.980 (95% confidence interval 0.963–0.996). **(F)** The Boruta feature selection method was used to identify genes predictive of disease outcome. Boruta method plot showing neutrophil genes that are associated with first DMSA outcome. 34 genes were classified as important, including *MAP2K2*, *FLG2*, *SVIP*, and *PGM2*. Confirmed relevant (green), tentative (yellow), and rejected (red) predictor boxplots are shown. Reference levels are shown in blue. **(G)** Receiver operating characteristic curve of a four-gene signature associated with APN, comparing first DMSA+ and DMSA– in acute RNA samples. The combined *MAP2K2*, *FLG2*, *SVIP*, and *PGM2* expression values were used, the resulting area under the curve was 0.944. **(H)** Boruta method plot showing no neutrophil gene significantly associated with second DMSA outcome. Eight interferon-related genes were classified as important, however, including *GBP6*, *HLA*, *IFNG*, and *IFNA*4.

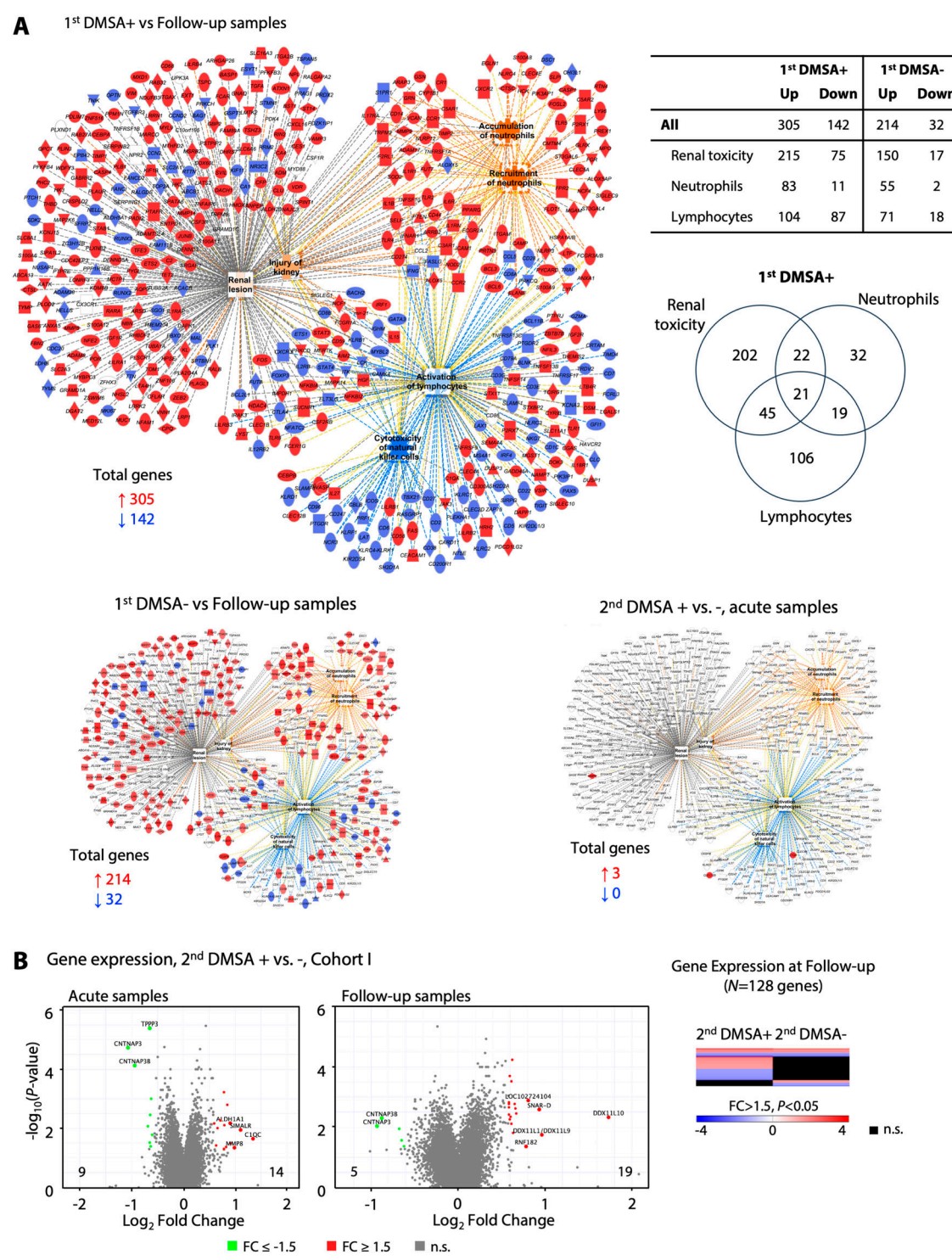

**Figure 5. Renal toxicity and gene expression in scarring patients.**
**(A)** Renal toxicity–associated gene network regulated in patients with APN compared with the follow-up samples. Renal toxicity genes were more strongly activated in patients with APN compared with febrile UTI (290 compared with 167 genes). Neutrophil-related genes were more strongly activated and lymphocyte-related genes more strongly inhibited. **(B)** Lack of renal scarring–specific gene expression at enrolment or at the 6-mo follow-up in patients with second DMSA+ (volcano plot directly comparing the second DMSA+ and second DMSA– groups). Lack of regulation of renal toxicity genes for the scarring-specific response.

the CXCR1 chemokine receptor was reduced in the susceptible families (Frendeus et al, 2000; Lundstedt et al, 2007). However, a comprehensive multi-omics approach has not previously been used

to study the molecular basis of APN susceptibility or disease severity in this patient group. In this study, exome genotyping and ORR analysis provided data suggesting that patients with APN (first

DMSA+) are genetically distinct from the first DMSA− group with febrile UTI, consistent with a genetic basis of susceptibility, potentially creating the immune imbalance detected here.

It is generally assumed that renal scars are caused by the most severe acute infections and that persistent inflammation is the essential scarring mechanism (Suárez-Álvarez et al, 2016). This study did not support this hypothesis, as there was no evidence that the most elevated acute response parameters predicted renal scarring. There was no evidence that the patients with a cytokine storm, elevated neutrophil degranulation or activated renal toxicity genes were the ones who subsequently developed renal scars. Instead, transcriptional activity and urine protein levels were low at the time of the second DMSA scan. These observations are interesting as they contradict the dogma in this field and suggest that mechanisms other than excessive acute inflammation may cause renal scarring. The study highlights the importance of studying the acute disease to understand its molecular determinants as well as long-term sequelae such as renal scarring.

## Materials and Methods

### Study design

Two prospective, analytical cohort studies were conducted to identify molecular susceptibility determinants and disease mechanisms in infants with their first febrile UTI episode. Patients were enrolled after written informed consent (Table S8), and a diagnosis of febrile UTI was confirmed by urine cultures and standard laboratory examinations (Figs S2 and S3). Patients with discordant urine culture results were excluded from the study.

APN was diagnosed by a positive first dimercaptosuccinic acid (DMSA) scan within 7 d of enrolment (Jakobsson et al, 1992). The patients were subjected to ultrasound examinations and VUR was detected by MCU. Renal scarring was detected by a positive second DMSA scan at follow-up after 6 mo (Ditchfield et al, 2002; Hoberman et al, 2003).

### Clinical cohorts

Cohort I included children older than 1 mo old admitted to the large tertiary teaching Hospital, KK Hospital in Singapore (n = 111, Table S1). Diagnosis and treatment followed an established clinical pathway and mandated routine tests included urinalysis, full blood count, CRP and blood culture (Fig S2). Inclusion criteria were a positive urine culture (single organism growth with counts > $10^4$ cfu/ml in catheterized urine or > $10^5$ cfu/ml in mid-stream clean-catch urine), temperature >38.5°C, pyuria and supported by CRP positivity (Xu et al, 2014). 88.7% of urine samples were collected from catheter and 11.3% were collected by mid-stream clean catch from older children where two samples were collected. Patients with discordant urine culture results were excluded from the study. Exclusion criteria were recurrent UTI, underlying renal or urological abnormalities known to patients or found on ultrasound of the kidney, urinary tract, and urinary bladder taken during admission. A total of 121 children were invited to participate in the study, eight declined further investigation, and DNA was not obtained from another two, resulting in a study cohort of 111 children (Table S1). Four patients with significant pathology requiring further actions were not included in the study (renal abscesses, pelviureteric junction obstruction with hydronephrosis, nephrocalcinosis, and duplex kidneys). There were seven cases of recurrent UTI, three patients had a recurrent UTI after 6 mo, and four had a recurrent UTI within 2 mo of the first UTI where the second DMSA was performed 4 mo after second UTI episode. Blood samples were obtained within 24 h of diagnosis for exome genotyping (n = 111) and whole genome transcriptomic analysis (n = 111), and urine samples for proteomic analysis. RNA and urine samples were further obtained after 6 mo.

Cohort II included children admitted to 29 pediatric centers in Sweden (n = 52, Tables S2 and S9). Diagnostic and treatment for febrile UTI followed an established clinical pathway; routine tests included urinalysis, full blood count, CRP, and blood cultures in severely ill children, before intravenous antibiotics is initiated (Fig S3 and Table S2) (Brandstrom & Lindén, 2021). Inclusion criteria were pyuria, a temperature >38.0°C and a positive urine culture. 86 children aged <1 yr with a clinical suspicion of a first episode of febrile UTI, were invited to participate in the study. Two patients with significant pathology requiring further actions were not included in the study (hydronephrosis, meatus stenosis, and balanic hypospadias). Blood samples were obtained for exome genotyping (n = 39) and whole genome transcriptomic analysis (n = 27) within 24 h of diagnosis. Urine and RNA samples were further obtained after 6 mo.

### Imaging studies

Patients were assigned a diagnosis of APN based on the outcome of the first dimercaptosuccinic acid (DMSA) renal scans, timed within 5–7 d of inclusion and defined as photopenic areas with preservation of the renal outline. A diagnosis of renal scarring was based on the outcome of a second DMSA scan after 6 mo. $^{99m}$Tc-DMSA was used as the radiopharmaceutical and imaging was with a dual-head camera with low-energy high-resolution collimators. Well-hydrated patients underwent DMSA scans using standard protocol for static renal study (Piepsz et al, 2001; De Palma, 2020).

Cohort I was recruited at a single center, all images were read by trained radiologists followed by vetting and final approval from an experienced senior radiologist designated for the study. Renal ultrasounds (US) were performed within 5 d and MCU 3–4 wk later.

Cohort II was recruited from a multicenter study; DMSA scans were performed within 7 d of inclusion, vetted, and evaluated locally by trained radiologists. All patients were examined by ultrasound and MCU was performed on selected children with dilated renal pelvis or reduced differential function on DMSA to detect VUR (Garin, 2019).

### Gene expression analysis

RNA was stabilized and purified from peripheral blood using Tempus blood RNA tubes and purification kit (Applied Biosystems),

collected at the time of diagnosis and 6-mo post-infection. RNA was subjected to expression microarray analysis: 100 ng of total RNA was amplified using the Affymetrix WT PLUS Reagent Kit and hybridized using the GeneTitan system onto GeneChip Human Gene 2.1 ST Arrays with probe sets measuring the expression of 72,688 transcripts, including a large number of non-coding and hypothetical transcripts. RNA samples collected during follow-up visits post-infection from the same study cohort were used as controls.

To ensure that all arrays were of high quality and suitable for downstream analysis, stringent quality control (QC) was applied to all arrays using NUSE (Normalized Unscaled SEs), RLE (Relative Log Expression), and MDS-analysis (Fig S16) as implemented in the statistical scripting language R and available from the Bioconductor repository (package oligo). NUSE analysis identified samples that had a median normalize unscaled error > 5% compared with the median of the other samples; the same samples also had suspicious patterns in their relative log expression and deviated significantly from the rest of the samples in the MDS plot. In total, seven samples failed QC and were excluded from further analysis. Clinical analysis did not detect any obvious anomalies in the corresponding patients.

Transcriptomic data were normalized using the Robust Multi-Chip Analysis algorithm implemented in the Transcriptome Analysis Console software (TAC v.4.0.1.36, Applied Biosystems; Thermo Fisher Scientific). The TAC software calls the limma differential expression portion of the Bioconductor package to provide fold change. Fold change was calculated by comparing each group to RNA obtained 6 mo after enrolment from patients with febrile UTI without renal involvement (acute DMSA−). Relative expression was analyzed by ANOVA using the empirical Bayes (eBayes) method, and Benjamini-Hochberg step-Up FDR-controlling procedure at alpha 0.05 to correct for multiple comparisons (Benjamini & Hochberg, 1995). Genes with a $P$-value < 0.05, an FDR adjusted $P$-value < 0.05 and an absolute fold change > 1.5 were considered differentially expressed. Heatmaps were constructed using the Gitools 2.1.1 software. Differentially expressed genes and regulated pathways were analyzed using Ingenuity Pathway Analysis software (QIAGEN), using right-tailed Fisher's Exact test followed by Benjamini-Hochberg correction for multiple testing.

### Direct comparisons

For direct comparison of RNA samples, analyses were carried out using the language and environment for statistical computing (R Core Team, 2013). Differentially expressed genes were identified using an empirical Bayes adjusted $t$ test, using the Limma framework (Ritchie et al, 2015) in R/Bioconductor, and the overall expression profile was visualized by MDS plots.

For the acute samples, 15,827 probe sets were identified as differentially expressed, indicating a profound transcriptional response (FDR-corrected $P$-value <0.05). Group differences were further analyzed using Generally Applicable Gene-Set Enrichment methodology to interrogate 3,105 pathways from nine different sources including Msigdb, Reactome, and Panther. In addition, RNA samples obtained at enrolment were examined for associations with first DMSA positivity using Gene Set Enrichment Analysis with the Reactome directory as database for pathways (Gillespie et al, 2022).

RNA samples were obtained at the 6-mo follow-up visit and gene expression was compared between patients who developed renal scarring and those where the acute changes had resolved (second DMSA+ compared with DMSA−). No significant differentially expressed genes were obtained after $P$-value adjustment.

The Boruta algorithm (v. 8.0) (Kursa & Rudnicki, 2010), a wrapper based random forest classification method, was used to identify important and unimportant genes using normalized expression data for each dataset. Genes included in the Neutrophil Degranulation (R-HSA-6798695) Reactome pathway and neutrophil biology genes compiled by Naranbhai et al (2015), or the Interferon alpha/beta Signaling (R-HSA-909733) and the Interferon Signaling (R-HSA-913531) Reactome pathways, were used to perform the analysis. Boruta first adds randomness to the given data set by creating shuffled copies of all features, thus creating Shadow Features. Then, it trains a random forest classifier on this extended data set (original and shadow attributes) and applies a feature importance measure, evaluating the importance of each feature. Finally, the Boruta Algorithm stops when all features are confirmed or rejected.

### Urine analysis

Urine samples from Cohort I were thawed, spun down, and supernatants were analyzed using Luminex multiplexed, bead-based kits: Human Cytokine Panel 1 and 2, Human Soluble Cytokine Receptor Panel, Kidney Injury Panel 3 and 4 (Merck Millipore), for a total of 84 measured proteins. Kits were processed as per manufacturer's protocols and read on the Flexmap 3D system (Luminex). Data were analyzed using Bio-Plex Manager 6.0 Software (Bio-Rad) with a 5-parameter curve-fitting algorithm applied for standard curve calculations. Urine CD177 levels were quantified using the Human CD177 ELISA Kit (#EH80RB; Invitrogen), as per the manufacturer's protocol. Significance was analyzed using the Mann–Whitney or Wilcoxon matched-pairs signed rank tests.

### Experimental kidney infection model

$Irf3^{-/-}$ mice on a C57BL/6 background were kindly provided by the Riken Bioresource Center, Japan, with permission from T Taniguchi (Sato et al, 2000). Mice were bred at Lund University, BMC animal facility, Lund, Sweden, and housed in specific pathogen–free individually ventilated cages (IVCs) at a constant temperature of 23°C on a 12-h light–dark cycle with lights on at 7:00 AM and ad libitum access to food and water. Female $Irf3^{-/-}$ mice were used for experiments at 9–15 wk of age.

After anesthesia by intraperitoneal (i.p.) injection of a cocktail of xylazine (0.22 mg; Vetmedic) and ketamine (1.48 mg; Intervet) in 100 $\mu$l of 0.9% NaCl solution, mice were infected with the prototype APN strain $E.$ $coli$ CFT073 (O6:K2:H1) (Mobley et al, 1990), $10^8$ CFU in 50 $\mu$l by intravesical inoculation through a soft polyethylene catheter. Before infection, $E.$ $coli$ CFT073 was cultured on tryptic soy agar (TSA) plates at 37°C for 16 h, harvested in PBS (pH 7.2) and diluted as appropriate in PBS.

Urine samples were collected from each mouse prior and at regular times after infection (24 h, 3 d, 5 d, and 7 d). Viable bacteria

counts were determined by growth on TSA plates (37°C, overnight) after appropriate serial dilutions. Neutrophil counts were determined from uncentrifuged urine observed using a hemocytometer chamber. Mice were euthanized by an overdose of isoflurane anesthesia (Dechra) after 24 h (day 1) and 7 d (day 7). Kidneys were aseptically removed, quickly frozen for subsequent RNA extraction, or embedded in O.C.T. compound for immunostaining.

### Whole genome transcriptome analysis of kidney tissue

Total RNA was extracted from murine kidneys with the RNeasy Mini Kit (QIAGEN) after disruption in RLT buffer supplemented with $\beta$-mercaptoethanol (1%) using Precellys Lysing kits (Bertin Technologies) and Tissuelyser (QIAGEN). Exactly 100 ng of total RNA was amplified and fragmented using the Affymetrix WT PLUS Reagent, followed by hybridization onto Clariom S Mouse HT (Affymetrix) for 16 h at 45°C. Then, the sample was washed, stained using the Affymetrix fluidic station 450 as per manufacturer's instruction (Hybridization, Washing, and Staining kit; Affymetrix). Microarrays were immediately scanned using the GeneTitan sytem (Affymetrix). Data were normalized using Robust Multi Average implemented in Transcriptome Analysis Console software (v.4.0.1.36, Applied Biosystems; Thermo Fisher Scientific). Relative expression was analyzed by ANOVA using the empirical Bayes method, and genes with an absolute fold change > 2.0 and P < 0.05 were considered differentially expressed. Heatmap was constructed in GraphPad Prism 10.

### Immunohistochemistry

Tissues embedded and frozen in O.C.T. were cryosectioned (8 $\mu$m thick sections; Leica microtome), collected on positively charged microscope slides (Superfrost/Plus; Thermo Fisher Scientific), fixed in acetone-methanol (1:1, 10 min), permeabilized (0.2% Triton X-100, 5% normal goat serum/PBS), and incubated overnight at 4°C with primary rat anti-neutrophil [NIMP-R14] (1: 200; ab2557; Abcam), rabbit anti-*E. coli* (1:100, NB200-579; BD bioscience), or rabbit polyclonal anti-CD177 (1:200; PA598759; Invitrogen) antibodies, followed by incubation for 1 h at room temperature with Alexa 488- or Alexa 568-labeled rabbit anti-rat and goat anti-rabbit IgG secondary antibodies (1:100, Molecular Probes, A-21210, A-11001 and A-11011). Nuclei were counterstained with DAPI (4',6-diamidino-2-phenylindole; 0.05 mM; Sigma-Aldrich) for 15 min at room temperature. Slides were examined by fluorescence microscopy (AX60; Olympus Optical), and fluorescence staining quantification was performed using Fiji (ImageJ).

### Exome genotyping and whole genome sequencing

Exome genotyping was performed using Illumina Infinium Exome beadchip technology that includes >240,000 markers from diverse populations and enriched with an additional 30,000 single nucleotide variants obtained from exome sequencing of Asian populations (Illumina Exome Asian 30 K chip), as previously described (Dunstan et al, 2014; Liu et al, 2017). For whole genome sequencing, a sequencing library was prepared using the TruSeq DNA library prep kit (Illumina) according to the manufacturer's instructions. This was sequenced on the Illumina NovaSeq 6000 platform, using S4 flow cell with 2 × 151-bp (paired end) reads, resulting in a HiSeq 30x coverage.

For Cohort I, DNA was from peripheral blood samples obtained at the time of diagnosis and stored at –80°C until extraction. For Cohort II, capillary blood was collected in $K_2$EDTA tubes (BD Microtainer MAP), and DNA was extracted using the QIAsymphony DNA Mini Kit (QIAGEN). Genomic DNA samples of at least 50 ng/$\mu$l concentration were amplified and enzyme digested as per manufacturer's instructions and further hybridized into the Illumina Infinium Exome beadchip.

For Cohort I, SNPs markers were called using the Illumina Genome Studio platform, where a training set of markers with 100% genotyping completion rate was applied. Genotyping data were corrected for strand issues using Genotype Harmonizer using the 1,000 Genomes data for the Southern Han Chinese (CHS) population, reducing data from 51,000 to 35,000 variants (Deelen et al, 2014). The corrected genotype data were then associated with the clinical endpoints using PLINK based on a genotypic model. Samples were pre-processed and quality controlled to exclude monomorphic sites, leaving 35,008 SNPs with at least one alternative allele. The average call rate across the entire cohort was >99.8% and the minor allele frequency was 20.3%.

Genotype differences between patient groups were analyzed by computing ORRs for the acute DMSA endpoint (Tables S6 and S7) and follow-up DMSA endpoint. The $log_2$ ORRs were calculated for DMSA positive and negative groups using the following formula: $log2\,ORRG_i = log2\left(\frac{P_i/N_i}{(P_j + P_k)/(N_j + N_k)}\right)$, where P and N represent the number of observed genotypes of individuals belonging to DMSA positive (P) and negative (N) groups, respectively, and i, j and k represent the three possible genotypes (G) observed at a genomic locus.

For Cohort II, Fisher´s exact test was applied to obtain P-values for the $log_2$ ORRs. Both analyses were performed using R (R Core Team, 2013), variants were filtered and P-values less than 0.01 were considered significant (Duggal et al, 2008; Fadista et al, 2016; Kaler & Purcell, 2019). The heatmap was prepared using pheatmap R-package (Kolde, 2015). The samples were ordered based on DMSA status and columns based on mean $log_2$ ORRs calculated for the DMSA positive group.

The genotype counts obtained for cohorts I and II were compared against the genotype counts obtained from the full gnomAD database. Only variants present in the gnomAD database were included in the analysis. The P-value for the genotype with the highest ORR was selected for each variant and annotated using Ensembl Variant Effect Predictor (McLaren et al, 2016). For each gene, the most significant variant (lowest P-value) was then selected as the representative variant. Finally, each individual was assigned an ORR, based on their genotype for the representative variant.

For Cohort I, clustering visualization was conducted using tSNE analysis. P-values were adjusted for multiple comparisons, and data were subjected to significance filtering (P-value and Q-value < 0.05). Significant representative variants were loaded individually in a heatmap representing a matrix for all individual patient genotype.

### Bacterial genome sequencing and fimbrial function

For each *E. coli* isolate, genomic DNA from an overnight culture in Luria broth (LB) was sheared to ~300 bp using a focused ultrasonicator (Covaris). A sequencing library was prepared using the TruSeq DNA library prep kit (Illumina) according to the manufacturer's instructions. This was sequenced using an Illumina HiSeq2000 platform with 2 × 76-bp or 2 × 101-bp reads. Phylogroup and virulence genes were called directly from the resulting fastq files using SRST (v0.1.8) (Inouye et al, 2014). Phylogroups were called using a custom database implementing the Clermont PCR-based system (Clermont et al, 2000) formatted for SRST2. Virulence factors (including fim, pap, and hly genes) were called using the VFDB database (Chen et al, 2005) as described in the SRST2 documentation. For validation of virulence factor calls, a BLAST-based method implemented with custom scripts was used on the assemblies. Assemblies for this validation were performed using Velvet version 1.2.10 (Zerbino & Birney, 2008) with a minimum contig cutoff of 500 bp, scaffolded with OPERA version 1.4.1 (Gao et al, 2011), and finished with FinIS version 0.3 (Gao et al, 2012).

The expression of functional fimbriae was evaluated by agglutination of erythrocytes expressing appropriate receptors (Kallenius et al, 1980; Leffler & Svanborg-Edén, 1980). For detection of P fimbriae, clinical *E. coli* isolates were grown overnight on TSA plates, and for type 1 fimbriae in LB (37°C, 18 h). Human $A_1P_1$ erythrocytes were used for detection of P fimbriae and guinea pig erythrocytes for detection of type 1 fimbriae. Erythrocytes were resuspended in PBS with or without the addition of 2.5% $\alpha$-methyl-D-mannopyranoside and mixed with a bacterial suspension on microscopy slides. Agglutination was documented as the aggregation of erythrocytes and clearance of the surrounding media.

### Study approvals

Ethical approval for Cohort I was obtained from SingHealth Centralized Institutional Review Board. Written, informed consent was obtained with adherence to the ICH guideline for Good Clinical Practice and all clinical investigators were qualified with certification in human research under the Collaborative Institutional Training Initiative (CITI) program. Ethical approval for Cohort II was obtained from the Swedish Ethical Review Authority (DNR 2015/884, 2016/799, 2017/164, and 2017/315). Written information was presented to the children´s caretakers and participation required a signed informed consent. Ethical approval for animal experiments was obtained from the Malmö/Lund Animal Experimental Ethics Committee at the Lund District Court, Sweden (#M119-16 and 6551-2021). Animal care and experimental protocols followed institutional, national, and European Union guidelines and were governed by the European Parliament and Council Directive (2016/63, EU), the Swedish Animal Welfare Act (Djurskyddslagen 1988:534), the Swedish Welfare Ordinance (Djurskyddsförordningen 1988:539) and Institutional Animal Care and Use Committee (IACUC) Guidelines. Results were reported in accordance with ARRIVE guidelines (https://arriveguidelines.org).

### Statistical analysis

Outcomes of enrolment and diagnostic procedures were evaluated using the SPSS program. Exome genotyping data were evaluated by computing ORR relative to DMSA outcome and visualized using the tSNE analysis. Gene expression and genotype associations were tested using linear regression and ANOVA models, as well as non-linear techniques including generalized linear- and mixed models. *P*-values were adjusted with multiple tests correction for big data using the Benjamini-Hochberg method, and data were subjected to significance filtering (*P*-value and Q-value < 0.05). Proteomics data were analyzed, after $\log_{10}$ transformation of concentration values, using two-way ANOVA with Šídák's multiple comparisons test. Immunohistology staining intensities were tested for normal distribution using the Shapiro-Wilk test, and differences compared with controls were analyzed by ordinary one-way ANOVA followed by Dunnett's multiple comparison.

*P*-values < 0.05 were considered significant. The use of specific biostatistics- and bioinformatics methods is further detailed in the Results and Supplementary Material.

## Data Availability

Genotyping and gene expression data used in this study are available in the supplemental material. Other supporting data sets can be made available from the corresponding author upon reasonable request.

## Supplementary Information

## Acknowledgements

The authors gratefully acknowledge the analytical support of Jane Pulman and Tommi Rantapero at Genevia Technologies Oy, Finland. The authors also thank the clinical study teams who participated in the two clinical studies. The clinical investigators involved in the Swedish Infant UTI study are listed in Table S9. The study was supported by grants from the Biomedical Research Council at the Agency for Science, Technology and Research Singapore (SIgN 09-025), the National Medical Research Council Singapore (#09-045), the Swedish Medical Research Council, European Research Council INFECT-ERA II program, Swedish Cancer Society, Österlund Foundation, Royal Physiographic Society, and HJ Forssman Foundation for Medical Research. The laboratory infrastructure was further supported by the European Union's Horizon 2020 research and innovation program under grant agreement No 954360, and by a grant from Hamlet BioPharma.

### Author Contributions

I Ambite: data curation, formal analysis, investigation, and writing—original draft, review, and editing.
SM Chao: data curation, writing—original draft, review and editing, and clinical investigations.

T Rosenblad: data curation, writing—original draft, and clinical investigations.

R Hopkins: formal analysis and investigation.

P Storm: formal analysis and investigation.

YH Ng: clinical investigations.

I Ganesan: clinical investigations.

M Lindén: data curation and clinical investigations.

F Haq: data curation and formal analysis.

TH Tran: formal analysis and investigation.

S Ahmadi: investigation.

B Lee: formal analysis.

SL Chen: data curation and formal analysis.

G Godaly: data curation and supervision.

P Brandström: supervision and clinical investigations.

JE Connolly: supervision and methodology.

C Svanborg: conceptualization, formal analysis, supervision, methodology, and writing—original draft, review, and editing.

## Conflict of Interest Statement

Funding for this study was exclusively academic. I Ambite, TH Tran, G Godaly, and C Svanborg are shareholders of Hamlet Biopharma, a biotech start-up company developing alternatives to antibiotics relevant to UTI treatment. I Ambite, S Ahmadi, G Godaly, and C Svanborg are part-time employees of Hamlet Biopharma. Patents for immunotherapy in UTI have been filed, with the scientists as inventors.

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
