## [Reviewer comments · Life Science Alliance]

Life Science Alliance

Molecular analysis of acute pyelonephritis - excessive innate and attenuated adaptive immunity

Ines Ambite, Sing Ming Chao, Therese Rosenblad, Richard Hopkins, Petter Storm, Yong Hong Ng, Indra Ganesan, Magnus Lindén, Farhan Haq, Hien Tran, Shahram Ahmadi, Bernett Lee, Swaine Chen, Gabriela Godaly, Per Brandström, John Connolly, and Catharina Svanborg

DOI: <https://doi.org/10.26508/lsa.202402926>

Corresponding author(s): Catharina Svanborg, Lund University

Review Timeline:

Submission Date:	2024-07-03
Editorial Decision:	2024-08-09
Revision Received:	2024-11-08
Editorial Decision:	2024-11-27
Revision Received:	2024-11-29
Accepted:	2024-11-29

Transaction Report:

August 9, 2024

Re: Life Science Alliance manuscript #LSA-2024-02926

Dr Catharina Svanborg
Institute of Laboratory Medicine
Section of MIG
Sölvegatan 23
Lund, Skane 22362
Sweden

Dear Dr. Svanborg,

Thank you for submitting your manuscript entitled "Excessive innate immune activation and neutrophil degranulation characterize acute pyelonephritis" to Life Science Alliance. The manuscript was assessed by expert reviewers, whose comments are appended to this letter. We invite you to submit a revised manuscript addressing the Reviewer comments.

Thank you for this interesting contribution to Life Science Alliance. We are looking forward to receiving your revised manuscript.

Sincerely,

B. MANUSCRIPT ORGANIZATION AND FORMATTING:

Reviewer #1 (Comments to the Authors (Required)):

Summary of the Work

The expansive study by Catharine Svanborg group titled "Excessive innate immune activation and neutrophil degranulation characterize acute pyelonephritis" describes comprehensive transcriptomics and proteomic analysis of 2 patient cohorts, Singapore cohort I, (n=111) and Swedish cohort II, (n=52) that were identified as febrile UTI patients (urine culture+, high fever, pyuria and increased CRP) and as APN using DMSA+ scan at the enrollment. The gene expression analysis was conducted using whole blood RNA and proteomics was done in urine samples. Gene expression was further characterized in patients that were DMSA+ at 6 month follow up and characterized as renal scarring subjects. This study identifies neutrophil degranulation as the major activated pathway and adaptive immune inhibition as major inhibited pathway in both cohorts. In cohort I, 1048 genes were upregulated and 438 down while in cohort II, 1247 were upregulated and 834 were down regulated in all patients with febrile UTI in both cohorts. When gene expression was compared between DMSA+ vs DMSA- it was found to be upregulated in DMSA+ subjects. Pathway analysis using IPA showed neutrophil degranulation, pathogen induced cytokine storm and phagosome formation as major activated pathways while T cell signaling, NF- κ B signaling as major inhibited pathways. Urine proteomic study was done using Luminex multiplex assay for 84 cytokine panels which found IL-8/CXCL8, IP-10, IL-6, MCP-1 among others as upregulated cytokine in acute vs follow up and in DMSA+ vs DMSA- patients. Then they went on to use sophisticated statistical tool (Boruta feature selection) to predict APN vs febrile UTI and found that neutrophil degranulation genes can predict between these 2 disease state. Study was well conducted and manuscript was well written.

Major Point:

Although neutrophil degranulation pathway was shown as the highest activated pathway by the authors using transcriptomics (Fig 2a, 2d), in proteomic analysis (Fig 3b), a M1 macrophage polarizing and proinflammatory protein MCP-1/CCL2 (Martha Gschwandtner et al, Front Immunol Review 2019, Kanda Hajime et al, JCI, 2006, Laura Beth Moore et al, Acta Biomater, 2015), (at #4 position) is ~4 fold higher in 1st DMSA+ vs DMSA- patients while IL-8 is 1.5 fold higher. In addition supplementary data Figure S9, a M1 macrophage inducing protein Osteopontin (OPN) which promotes fibrosis (Zhihao Xu et al, J Clin Transl Hepatol, 2023) is 3.41 fold higher in 2nd DMSA+ patients. This suggest that M1 polarizing macrophage residing in kidney may be playing critical role in scarring phenotype. Given the important role of renal resident macrophages in renal scarring in C3H/HeOuj strain (Li et al, AJP Renal Physiol, 2017) and their role in cystitis (Mariano L L et al, Sci Adv, 2020) and in undermining adaptive immunity (Gabriela Mora-Bau, Plos Pathog, 2015). In light of these important findings role of macrophages need to be at least discussed. Also the role of collecting duct cells (ICs) which play important role in countering UPEC strain CFT073 in this setting need to be discussed.

To study if neutrophil degranulation and CD177 are affected by kidney infection, IRF3^{-/-} mouse was used which authors have shown previously its role in destructive innate immunity, another genetically APN susceptible and scarring phenotype strain such as C3H/HeOuj strain may be useful. Also the staining for bacteria or neutrophil is not clear in spite of negative control shows not staining. What is the location of bacteria in kidney.

Minor Points:

Figure 1 and Figure 2 file labels are reversed.

Fig S2 Clinical pathway cohort II figure is not clear

Clarify when all Febrile UTI patients were compared and when only DMSA+ were compared.

Reviewer #2 (Comments to the Authors (Required)):

Ambite and colleagues in this manuscript report the results of experiments (mostly peripheral blood RNA sequencing) performed with human subjects - infants presenting with febrile UTI and the subset of these that have acute pyelonephritis (APN) as defined by an acute-stage DMSA scan. The results are interesting, it is a type of analysis on a sizable cohort of human febrile UTI (fUTI) patients that hasn't been done before, and the supplementary data are voluminous. That said, the RNA sequencing results substantially reflect the peripheral neutrophilia that would be expected to characterize these patients. Furthermore, the descriptors used to convey the magnitude or significance of the changes are overstated throughout the paper, as outlined in

more detail below. Suggestions for improvement of the manuscript follow.

Major points in terms of additional analysis:

1. Was there analysis of the allelic variation in terms of the prediction of whether these gene variants would be damaging to the proteins they encode?
2. Were any sex differences noted in the peripheral blood gene expression profiles?

Major points in terms of writing and presentation:

1. There is overuse of the descriptors "hyper," "excessive," and such throughout the paper. As just one example, on line 203 it would be accurate to say that the results identify activation of innate immune responses, not "hyper-activated." In a patient with any febrile bacterial infection, the peripheral blood would, in fact, be expected to reflect neutrophilia and innate immune activation. In such subjects, whole-blood RNA sequencing would be expected to reflect a comparatively high population of neutrophils (and therefore transcripts of neutrophil genes).
2. Similarly, "cytokine storm" happens to be included in the name of a KEGG-type gene pathway (line 173), but the use of this terminology in the paper should be limited to that context. These human subjects did not have what a clinician would term cytokine storm syndrome. Invoking a "local cytokine storm in the urinary tract" (lines 209-222) is an over-description of what is in fact an expected innate immune activation. Other examples where the "hyper"-type descriptors need to be removed or toned down are in the title, the abstract (lines 50-51 "an extreme hyper-inflammatory disorder" resembling "cytokine storms"), and in the main text lines 203, 204, 217, 221, 222, 224, 279, 333, 334, and 337.
3. There are multiple uses of the terms "predictive" and "predictor" that are not appropriate beginning at line 239. The first DMSA scan was used to classify patients into the APN or fUTI groups; one cannot then say that a group of 34 genes (line 244) or any subset of these (line 246) "predicts" APN. Instead, transcription of these genes associates with the phenotype as defined by the first DMSA scan. Other examples where the "predictive" terms should be altered are in Discussion lines 369, 371, and 372. (Of note, the observation that there were no acute peripheral blood gene signatures that "predict" renal scarring [e.g., lines 249-250] is accurate - here, you have a temporal relationship between the early-stage immune response and what is being "predicted" - later scarring.)

Minor points:

1. In the first Results paragraph, an additional sentence describing the two cohorts (and why there are two) would help readability.
2. In the paragraph starting on line 160, it seems that "the group with febrile UTI" (line 165) means the cohort members who had a negative first DMSA scan, though this should be made clear for the reader.
3. There are no call-outs in the Results text to Figures 3b and 3c.
4. In the Discussion paragraph regarding the potential clinical utility of the authors' findings (starting on line 369), it should be explained that even though CD177 was found to be elevated in the fUTI subjects, it is very likely that any significant bacterial infection that causes peripheral neutrophilia (pneumonia, appendicitis...) would likely also elevate CD177 transcript in the peripheral blood; in other words, it would be expected to have poor specificity for fUTI.
5. Fig S4e shows the overall proportion of strains carrying genes from the hlyCABD operon, but were there differences between the DMSA1-positive and negative groups? (similar to the analysis done for fim and pap in Fig S4c)
6. Discussion, lines 338-339 - what is meant by "potential failure of adaptive immunity to cope with large numbers of dysfunctional inflammatory cells"? This assertion is nonspecific and quite speculative, and again posits that the myeloid cells are "dysfunctional" rather than executing their normal functions.
7. The Discussion paragraph starting on line 356 is not topically focused. It starts out talking about the DNA variants in the two cohorts and finishes talking about how the RNA profiles in the acute phase didn't predict scarring.

Reviewer #3 (Comments to the Authors (Required)):

This is an important paper combining sound epidemiological/clinical principles of study design and strong basic science work. However, there is undue emphasis on the comparisons between baseline (when the child has a major systemic infection) and follow-up (when the child is absolutely well). Inferences made by examining comparisons between sick and well state dominate the narrative. The real contribution of this study is that children underwent DMSA scanning (twice) to determine their true outcome, and it is this that should be highlighted. I respectfully submit the following specific critiques in hopes of improving the final manuscript.

- 1) In all figures and tables, would emphasize the comparison of DMSA + vs - (rather than acute vs. follow-up). Recommend moving results for the comparison relating to acute vs follow-up to the supplement.
- 2) Would rewrite the results, discussion, conclusion, and abstract, emphasizing findings related the DMSA + vs - comparison.
- 3) The authors conclude that APN can be characterized as "cytokine storm," that there is "excessive" immune system activation, or that UTIs are a "hyper-inflammatory" disorder. These statements need more justification. What results exactly (when looking at the DMSA + vs - comparisons) support this?
- 4) Inclusion criteria say that the study required an elevated CRP but the Tables suggest otherwise. Clarify the inclusion criteria.
- 5) How many of the urines were collected by catheter. For clean catch, were 2 samples required. What was done if the two were discordant?

- 6) What urinalysis findings, if any, were required for inclusion
- 7) Did one of the studies enroll 0-1 month olds?
- 8) What happened to children who had a recurrent before the 2nd DMSA?
- 9) I assume the radiologist was blinded to study results when they read the scans. If so, this should be stated.
- 10) Line 183. The term "regulated" is not well explained
- 11) Some of the proportions are presented without confidence interval (example on lines 319 and 321).
- 12) It appears that some of the analyses were not corrected for overfitting/multiple comparisons. For example, the tSNE algorithm is said to have correctly distinguished those with scars from those without with an accuracy of 99.1%.
- 13) Figure 2. The main findings of interest are 2d and e. The rest of the parts, as mention earlier, add only confusion
- 14) Figure 3. Only 3c seems relevant but it is not clear how this was obtained, what the table vs. figure are supposed to convey, and whether it add to the results presented in the other tables/figures
- 15) Figure 4. 4b shows that the same genes were expressed by both DMSA + and - children. But the title of the figure states that neutrophil-related genes predict APN. This seems inconsistent. 4c: there is no legend to explain what the colors or asterisks mean. Drop 4d, e. Instead emphasize 4f,g, and h. The word "additional" in the figure is also confusing. In addition to "importance", it seems important to present FDR corrected p values and FC for the neutrophil genes that differ in DMSA + and negative children.
- 16) Figure 5 shows that 3 genes differed in follow-up DMSA + vs -. Does not seem that this merits a figure
- 17) Figure s4d middle panel (Guinea pig) is confusing.
- 18) Using the term proteomics to refer to a targeted LUMINEX panel seems inappropriate. Were the p values corrected for multiple comparisons for the cytokines?
- 19) Was the immune response different in very young infants?
- 20) What are "renal toxicity" genes? Clarification on the methods used for this pathway analysis are needed.
- 21) The abstract mentioned attenuation of the adaptive immune response. This is not highlighted in the figures and results sufficiently. What comparison between DMSA + and negative children supports this statement? Add a figure showing this
- 22) The abstract states that: neutrophil degranulation genes were disease associated. This is vague. What disease? UTI? APN? Scarring? What is the function of the 4 genes that were the most important?

Reviewer #1 (Comments to the Authors (Required)):**Summary of the Work**

The expansive study by Catharine Svanborg group titled "Excessive innate immune activation and neutrophil degranulation characterize acute pyelonephritis" describes comprehensive transcriptomics and proteomic analysis of 2 patient cohorts, Singapore cohort I, (n=111) and Swedish cohort II, (n=52) that were identified as febrile UTI patients (urine culture+, high fever, pyuria and increased CRP) and as APN using DMSA+ scan at the enrollment. The gene expression analysis was conducted using whole blood RNA and proteomics was done in urine samples. Gene expression was further characterized in patients that were DMSA+ at 6 month follow up and characterized as renal scarring subjects. This study identifies neutrophil degranulation as the major activated pathway and adaptive immune inhibition as major inhibited pathway in both cohorts. In cohort I, 1048 genes were upregulated and 438 down while in cohort II, 1247 were upregulated and 834 were down regulated in all patients with febrile UTI in both cohorts. When gene expression was compared between DMSA+ vs DMSA- it was found to be upregulated in DMSA+ subjects. Pathway analysis using IPA showed neutrophil degranulation, pathogen induced cytokine storm and phagosome formation as major activated pathways while T cell signaling, NF-kB signaling as major inhibited pathways. Urine proteomic study was done using Luminex multiplex assay for 84 cytokine panels which found IL-8/CXCL8, IP-10, IL-6, MCP-1 among others as upregulated cytokine in acute vs follow up and in DMSA+ vs DMSA- patients. Then they went on to use sophisticated statistical tool (Boruta feature selection) to predict APN vs febrile UTI and found that neutrophil degranulation genes can predict between these 2 disease state. Study was well conducted and manuscript was well written.

We thank the reviewer for these positive comments. We realise from these comments that the paper appeared to focus on the strong neutrophil response and that the cytokine storm response was not made sufficiently clear. The results clearly show that acute pyelonephritis is characterised by a cytokine storm response systemically and in urine, with a similar profile as observed during COVID or sepsis. We have revised the presentation of the data to first emphasize the broad response and the cytokine storm and subsequently the specific networks, including the neutrophil degranulation response.

Major Point:

Although neutrophil degranulation pathway was shown as the highest activated pathway by the authors using transcriptomics (Fig 2a, 2d), in proteomic analysis (Fig 3b), a M1 macrophage polarizing and proinflammatory protein MCP-1/CCL2 (Martha Gschwandtner et al, Front Immunol Review 2019, Kanda Hajime et al, JCI, 2006, Laura Beth Moore et al, Acta Biomater, 2015), (at #4 position) is ~4 fold higher in 1st DMSA+ vs DMSA- patients while IL-8 is 1.5 fold higher. In addition supplementary

data Figure S9, a M1 macrophage inducing protein Osteopontin (OPN), which promotes fibrosis (Zhihao Xu et al, J Clin Transl Hepatol, 2023) is 3.41 fold higher in 2nd DMSA+ patients. This suggest that M1 polarizing macrophage residing in kidney may be playing critical role in scarring phenotype. Given the important role of renal resident macrophages in renal scarring in C3H/HeOuJ strain (Li et al, AJP Renal Physiol, 2017) and their role in cystitis (Mariano L L et al, Sci Adv, 2020) and in undermining adaptive immunity (Gabriela Mora-Bau, Plos Pathog, 2015). In light of these important findings role of macrophages need to be at least discussed. Also the role of collecting duct cells (ICs), which play important role in countering UPEC strain CFT073 in this setting need to be discussed.

We thank the reviewer for these comments and the suggested references and have included, as well as a comment on the data in the Results and the interpretation in the Discussion.

While OPN is a very interesting molecule with possible roles in renal pathology, we do not find that the OPN data stands out compared to the more strongly regulated markers.

To study if neutrophil degranulation and CD177 are affected by kidney infection, IRF3^{-/-} mouse was used which authors have shown previously its role in destructive innate immunity, another genetically APN susceptible and scarring phenotype strain such as C3H/HeOuJ strain may be useful. Also the staining for bacteria or neutrophil is not clear in spite of negative control shows not staining. What is the location of bacteria in kidney.

Text explaining the infection model has been added to the revised paper, including comments on neutrophils and bacteria. The images in Figure S10 show the presence of bacteria and neutrophils in the renal papilli.

The neutrophil response in infected kidneys has been extensively studied, as reviewed in the Discussion of the paper. Tlr4 normal mice are more responsive than the Tlr4 deficient C3H/HeJ mouse, as we have shown in many publications (Shahin,Hagberg). Disease progression has been studied in C3H/HeOuJ mice, with interesting results.

We switched to using C57BL/6 mice, to screen for single genes that define the severity of acute pyelonephritis in the murine model. These studies have confirmed the importance of Tlr4 signalling for the response and have added information on neutrophil genes and transcription factors genes that regulate innate immunity and disease severity. These genes are broad regulators of innate immunity and will certainly affect many different cell types at the site of infection and systemically. As far as we are aware, genetic variants depleted for specific cell types such as

macrophages or their activation markers are not available in the C3H/HeOuJ background.

Minor Points:

Figure 1 and Figure 2 file labels are reversed.

We apologise

Fig S2 Clinical pathway cohort II figure is not clear

Please see the publication of the Swedish study for clarification of the clinical pathway (Brandström, P, Linden, M. Acta Paediatrica. 2021; 110:1759–1771 and Linden, M et al. Pediatr Nephrol 2024; 39(11):3251-3262).

Clarify when all Febrile UTI patients were compared and when only DMSA+ were compared.

The DMSA status of the groups has been clarified in the text.

Reviewer #2 (Comments to the Authors (Required)):

Ambite and colleagues in this manuscript report the results of experiments (mostly peripheral blood RNA sequencing) performed with human subjects - infants presenting with febrile UTI and the subset of these that have acute pyelonephritis (APN) as defined by an acute-stage DMSA scan. The results are interesting, it is a type of analysis on a sizable cohort of human febrile UTI (fUTI) patients that hasn't been done before, and the supplementary data are voluminous. That said, the RNA sequencing results substantially reflect the peripheral neutrophilia that would be expected to characterize these patients. Furthermore, the descriptors used to convey the magnitude or significance of the changes are overstated throughout the paper, as outlined in more detail below. Suggestions for improvement of the manuscript follow.

We appreciate the comments about neutrophils and their importance for acute pyelonephritis. As stated above, previous studies have provided extensive mechanistic to support the role of neutrophils in acute pyelonephritis in the murine model. It is gratifying to now be able to confirm some of this biology with quantitative screening tools in human disease.

The predominance of neutrophil degranulation genes indicates that this part of the neutrophil life cycle is especially important for the disease response. We also show by proteomics technology that neutrophil chemoattractants and proinflammatory cytokines are very strongly upregulated locally in the urinary tract, confirming the local origin of the cytokine response and the extent to which this is activated by infection.

The disease severity is objectively documented as a cytokine storm locally in urine and systemically at the RNA level. These findings place infants with febrile UTI, in the same category as patients with other very severe infections such as COVID and sepsis.

As stated above, we have edited the paper to clarify these points.

Major points in terms of additional analysis:

1. Was there analysis of the allelic variation in terms of the prediction of whether these gene variants would be damaging to the proteins they encode?

As stated in the Methods, allelic variation was carefully considered during the DNA data analysis. We have reported overall differences in genetic profiles between patients with 1st or 2nd DMSA positivity. For each gene, the most significant variant (lowest P-value) was then selected as the representative variant. Finally, each individual was assigned an ORR, based on the representative variant.

We share with the reviewer the curiosity about additional genetic variation. For this paper, we have limited the analysis to the 67 genes that were polymorphic in both patient populations. There will obviously be many more interesting questions to address in the future, in this dataset.

2. Were any sex differences noted in the peripheral blood gene expression profiles?
Interesting.

Except for x and y linked genes, no obvious sex difference was observed in the gene expression profiles. PCA plots with gender information have been added to the Supplementary material.

Major points in terms of writing and presentation:

1. There is overuse of the descriptors "hyper," "excessive," and such throughout the paper. As just one example, on line 203 it would be accurate to say that the results identify activation of innate immune responses, not "hyper-activated." In a patient with any febrile bacterial infection, the peripheral blood would, in fact, be expected to reflect neutrophilia and innate immune activation. In such subjects, whole-blood RNA

sequencing would be expected to reflect a comparatively high population of neutrophils (and therefore transcripts of neutrophil genes).

We must emphasize that this is a very strong response and this has to be made clear. These infants have a cytokine storm locally in the urinary tract and systemically in circulating cells. This is hyper-activation of the innate immune response as a cause of acute symptoms and disease.

The text has been revised to accommodate the reviewer's concern. Regarding the wording, hyper-inflammatory disorders are an accepted concept. Out-of-control or run away immune responses are often discussed as well as abnormal inflammation.

2. Similarly, "cytokine storm" happens to be included in the name of a KEGG-type gene pathway (line 173), but the use of this terminology in the paper should be limited to that context. These human subjects did not have what a clinician would term cytokine storm syndrome. Invoking a "local cytokine storm in the urinary tract" (lines 209-222) is an over-description of what is in fact an expected innate immune activation. Other examples where the "hyper"-type descriptors need to be removed or toned down are in the title, the abstract (lines 50-51 "an extreme hyper-inflammatory disorder" resembling "cytokine storms"), and in the main text lines 203, 204, 217, 221, 222, 224, 279, 333, 334, and 337.

It is essential to learn from this study and accept that infants with febrile UTI have a cytokine storm, locally in the urinary tract and systemically in circulating cells. A similar pattern has been seen in mice that are genetically susceptible to APN (Irf3^{-/-} mice).

The response has been defined as a cytokine storm by Ingenuity Pathway Analysis, based on large number of activated genes. The KEGG database is included in IPA and we have not selected for any specific gene sets.

We have retained the cytokine storm wording whenever we refer to the actual data analysis, which specifies this outcome. We have also modified the language to conform with the hyper-inflammatory disorder definitions described above.

3. There are multiple uses of the terms "predictive" and "predictor" that are not appropriate beginning at line 239. The first DMSA scan was used to classify patients into the APN or fUTI groups; one cannot then say that a group of 34 genes (line 244) or any subset of these (line 246) "predicts" APN. Instead, transcription of these genes associates with the phenotype as defined by the first DMSA scan. Other examples where the "predictive" terms should be altered are in Discussion lines 369, 371, and 372. (Of note, the observation that there were no acute peripheral blood gene signatures that "predict" renal scarring [e.g., lines 249-250] is accurate - here,

you have a temporal relationship between the early-stage immune response and what is being "predicted" - later scarring.)

We thank the reviewer for these comments and have further revised the text to clarify these points, especially limiting the use of the word prediction to groups separated in time.

The reviewer refers to the Results text using the BORUTA analysis, which is a program designed to evaluate the predictive power of individual genes in complex data sets. There are two outcomes examined here; the severity of acute infection, distinguishing DMSA+ from DMSA- patients and renal scarring. The paper focuses on mechanisms of acute disease and determinants of acute disease severity and we must be at liberty to choose this end point.

We predict that it will become essential to use parameters of acute disease severity in the future, when antibiotics can no longer be the "drug of choice" and patients may be selected for alternative therapies, based on their disease response profile.

Minor points:

1. In the first Results paragraph, an additional sentence describing the two cohorts (and why there are two) would help readability.

Has been amended.

2. In the paragraph starting on line 160, it seems that "the group with febrile UTI" (line 165) means the cohort members who had a negative first DMSA scan, though this should be made clear for the reader.

Has been amended.

3. There are no call-outs in the Results text to Figures 3b and 3c.

The text has been carefully edited.

4. In the Discussion paragraph regarding the potential clinical utility of the authors' findings (starting on line 369), it should be explained that even though CD177 was found to be elevated in the fUTI subjects, it is very likely that any significant bacterial infection that causes peripheral neutrophilia (pneumonia, appendicitis...) would likely also elevate CD177 transcript in the peripheral blood; in other words, it would be expected to have poor specificity for fUTI.

We do not agree with the reviewer. Neutrophils are recruited to the site of infection by chemokines produced at the local infection site, at least initially. By proteomic analysis of urine, we tie the systemic findings to the local response. APN is an infection initiated in the kidneys, by uropathogenic E. coli, with a specific innate immune response profile.

Neutrophil responses to infection have different tissue origins, are caused by different pathogens locally and show different local innate immune response characteristics. The fact that neutrophil counts are elevated in many infections reflects the mechanisms of neutrophil mobilisation but the profile of the innate immune response differs between diseases and pathogens.

5. Fig S4e shows the overall proportion of strains carrying genes from the hlyCABD operon, but were there differences between the DMSA1-positive and negative groups? (similar to the analysis done for fim and pap in Fig S4c).

There was no statistical difference in virulence gene distribution between the DMSA+ and DMSA- groups. Only 28-30% of the strains were hly positive. Proportions comparing DMSA+ and – groups have been added.

6. Discussion, lines 338-339 - what is meant by "potential failure of adaptive immunity to cope with large numbers of dysfunctional inflammatory cells"? This assertion is nonspecific and quite speculative, and again posits that the myeloid cells are "dysfunctional" rather than executing their normal functions.

We understand the reviewer's concern and have revised the text. The inverse regulation of innate and adaptive immunity is still quite striking and has not previously been described.

7. The Discussion paragraph starting on line 356 is not topically focused. It starts out talking about the DNA variants in the two cohorts and finishes talking about how the RNA profiles in the acute phase didn't predict scarring.

Has been clarified and the discussion has been revised. The important conclusion is that the patients who develop renal scarring are a genetically distinct subset of the 1st DMSA+ group.

Reviewer #3 (Comments to the Authors (Required)):

This is an important paper combining sound epidemiological/clinical principles of study design and strong basic science work. However, there is undue emphasis on the comparisons between baseline (when the child has a major systemic infection) and follow-up (when the child is absolutely well). Inferences made by examining comparisons between sick and well state dominate the narrative. The real contribution of this study is that children underwent DMSA scanning (twice) to determine their true outcome, and it is this should be highlighted. I respectfully submit the following specific critiques in hopes of improving the final manuscript.

We thank the reviewer for the positive words but do not agree with the comments.

It appears ethically and medically questionable to disregard this severe acute infection, with characteristics of a cytokine storm. Would a patient not be of interest unless he/she develops renal scarring? The era of liberal antibiotic use is over, we must up our game to better distinguish and treat the acute infections.

There are many reasons for taking the trouble to do prospective studies with long-term follow up rather than collecting cross-sectional data only to study sequelae or chronicity. How can we understand the sequelae if we don't have the acute parameters to distinguish different patient subsets?

We clearly show that the acute disease is serious - very familiar to the paediatrician, who has the clinical algorithms but no laboratory data to support the diagnosis. CRP and fever are useful but clearly not sufficient.

1) In all figures and tables, would emphasize the comparison of DMSA + vs - (rather than acute vs. follow-up). Recommend moving results for the comparison relating to acute vs follow-up to the supplement.

The terminology DMSA + vs - is now used throughout the paper in response to the reviewer's comments. The cytokine storm response in the 1st DMSA + patients is stronger than in the 1st DMSA - group.

The longitudinal perspective is absolutely essential, to conclude that the patients with the strongest acute response are not necessarily the ones who develop renal scars. This is a change of dogma. We would challenge the "lazy" focus only on the patients who develop renal scarring.

2) Would rewrite the results, discussion, conclusion, and abstract, emphasizing findings related the DMSA + vs - comparison.

The text was already written to emphasize this point but we have tried to clarify this further.

3) The authors conclude that APN can be characterized as "cytokine storm," that there is "excessive" immune system activation, or that UTIs are a "hyper-inflammatory" disorder. These statements need more justification. What results exactly (when looking at the DMSA + vs - comparisons) support this?

The data is clearly presented with P values defining differences between the groups that were compare, including the 1st DMSA+ versus - groups. Please see comment to reviewer 1 above. Yes, the cytokine storm response was higher in the 1st DMSA+ group.

4) Inclusion criteria say that the study required an elevated CRP but the Tables suggest otherwise. Clarify the inclusion criteria.

The patients with low CRP levels were mentioned in the Supplementary Tables and the range of CRP levels as shown in Figure 1. There were 5 patients (4.6%) with CRP levels ≤ 10 mg/L that were included in the study. The remaining clinical data supported a diagnosis of febrile UTI in these patients. Elevated CRP levels were recommended but low CRP levels were not an exclusion criterion.

5) How many of the urines were collected by catheter. For clean catch, were 2 samples required. What was done if the two were discordant?

In the Singapore study, 88.7% were catheterised urine samples (all of whom were infants requiring admission and urgent urine collected for culture and sensitivity before treatment). The remaining 11.3% were older children whose urine was collected by mid-stream clean catch. For clean catch, 2 samples were collected. Patients with discordant urine culture results were excluded from study.

6) What urinalysis findings, if any, were required for inclusion

Pyuria and bacteriuria of single organism with counts $>10,000$ cfu/ml for catheterised urine and counts $>100,000$ cfu/ml for clean catch.

7) Did one of the studies enroll 0-1 month olds?

In Cohort I patients were older than 1 month. Five patients in Cohort II were less than 1 month old. The patient included in the gene expression analysis was not an outlier.

8) What happened to children who had a recurrent before the 2nd DMSA?

There were 7 cases of recurrent UTI, 4 occurring within 2 months of 1st UTI with 2nd DMSA done 4 months after 2nd UTI and 3 patients had recurrent UTI after 6 months whose 2nd DMSA were already done by the time of 2nd UTI.

This information has been added in the revised paper.

9) I assume the radiologist was blinded to study results when they read the scans. If so, this should be stated.

Yes, has been stated.

10) Line 183. The term "regulated" is not well explained

Has been clarified. Regulated means differentially expressed compared to the follow up group - either activated or inhibited.

11) Some of the proportions are presented without confidence interval (example on lines 319 and 321).

The ORR data is given with a confidence interval. P values are given overall - not all data is of such a nature that confidence intervals are relevant.

12) It appears that some of the analyses were not corrected for overfitting/multiple comparisons. For example, the tSNE algorithm is said to have correctly distinguished those with scars from those without with an accuracy of 99.1%.

As stated in the manuscript, the data has been corrected for multiple comparisons. The variants identified as scarring-related showed specificity for the scarring group.

"t-distribution Stochastic Neighbor Embedding (tSNE) analysis was used for clustering visualization. P-values were adjusted for multiple comparisons and data was subjected to significance filtering (P-value and Q-value < 0.05)".

13) Figure 2. The main findings of interest are 2d and e. The rest of the parts, as mention earlier, add only confusion

As stated above, this study investigates, for the first time, the molecular basis of acute pyelonephritis with genome wide technology. This is not a scarring only study.

14) Figure 3. Only 3c seems relevant but it is not clear how this was obtained, what the table vs. figure are supposed to convey, and whether it add to the results presented in the other tables/figures

See response above. This is a comparison of urine proteomics between the 1st DMSA + and - groups.

15) Figure 4. 4b shows that the same genes were expressed by both DMSA + and - children. But the title of the figure states that neutrophil-related genes predict APN. This seems inconsistent. 4c: there is no legend to explain what the colors or asterisks mean. Drop 4d, e. Instead emphasize 4f,g, and h. The word "additional" in the figure is also confusing. In addition to "importance", it seems important to present FDR corrected p values and FC for the neutrophil genes that differ in DMSA + and negative children.

Has been clarified. The gene list is provided.

16) Figure 5 shows that 3 genes differed in follow-up DMSA + vs -. Does not seem that this merits a figure

This is the point. An amazing and totally unexpected finding.

17) Figure s4d middle panel (Guinea pig) is confusing.

Guinea pig blood is used to detect type 1 fimbriae by hemagglutination.

18) Using the term proteomics to refer to a targeted LUMINEX panel seems inappropriate. Were the p values corrected for multiple comparisons for the cytokines?

The analysis is a 2way ANOVA with Šídák's multiple comparisons test.

19) Was the immune response different in very young infants?

Age criteria were patients older than 1 month (that is excluding neonates).

20) What are "renal toxicity" genes? Clarification on the methods used for this pathway analysis are needed.

Additional data has been added. These data sets are curated by the IPA software, which selects for relevance to renal disease from a broad overview of the literature.

21) The abstract mentioned attenuation of the adaptive immune response. This is not highlighted in the figures and results sufficiently. What comparison between DMSA + and negative children supports this statement? Add a figure showing this

The results section and discussion of the revised paper now emphasize adaptive immunity.

22) The abstract states that: neutrophil degranulation genes were disease associated. This is vague. What disease? UTI? APN? Scarring? What is the function of the 4 genes that were the most important?

More information has been added to clarify the attenuation of the adaptive immune response in the DMSA+ group.

November 27, 2024

RE: Life Science Alliance Manuscript #LSA-2024-02926R

Catharina Svanborg
Institute of Laboratory Medicine
Section of MIG
Sölvegatan 23
Lund, Skane 22362
Sweden

Dear Dr. Svanborg,

Thank you for submitting your revised manuscript entitled "Molecular analysis of acute pyelonephritis - excessive innate and attenuated adaptive immunity". We would be happy to publish your paper in Life Science Alliance pending final revisions necessary to meet our formatting guidelines.

- please be sure that the authorship listing and order is correct
- please add ORCID ID for corresponding author-you should have received instructions on how to do so
- please add the Twitter handle of your host institute/organization as well as your own or/and one of the authors in our system
- please incorporate your supplementary methods into your main manuscript, and incorporate the supplemental References into the main Reference list
- please upload your supplemental figures as single files and add your supplemental figure legends to the main manuscript text
- please consult our manuscript preparation guidelines <https://www.life-science-alliance.org/manuscript-prep> and make sure your manuscript sections are in the correct order
- please use the [10 author names, et al.] format in your references (i.e. limit the author names to the first 10)
- please upload your supplementary table files as excel or doc files
- For figure S4, you have panels a-f in your figure, but the panel f is missing from the figure legend; please correct
- please add a figure callout for Figure 1a and Figure S16 and S17 to your main manuscript text

Figure Check:

- please add scale bars to Figure 1F

A. FINAL FILES:

-- Summary blurb (enter in submission system): A short text summarizing in a single sentence the study (max. 200 characters including spaces). This text is used in conjunction with the titles of papers, hence should be informative and complementary to the title. It should describe the context and significance of the findings for a general readership; it should be written in the

present tense and refer to the work in the third person. Author names should not be mentioned.

B. MANUSCRIPT ORGANIZATION AND FORMATTING:

Thank you for your attention to these final processing requirements. Please revise and format the manuscript and upload materials within 5 days.

Sincerely,

Reviewer #1 (Comments to the Authors (Required)):

Reviewer concerns adequately addressed.

Reviewer #2 (Comments to the Authors (Required)):

The authors have responded to all of the critiques offered by this reviewer. I do not have any further issues with the data and agree this is an interesting study. There remains some difference of opinion about some of the verbiage used in the interpretation of the data (as reflected in the initial comments), but this is now an editorial matter and should not ultimately preclude publication.

November 29, 2024

RE: Life Science Alliance Manuscript #LSA-2024-02926RR

Prof. Catharina Svanborg
Lund University
Laboratory Medicine, MIG
Klinikgatan 28
BMC B13
Lund, Skane 22242
Sweden

Dear Dr. Svanborg,

Thank you for submitting your Research Article entitled "Molecular analysis of acute pyelonephritis - excessive innate and attenuated adaptive immunity". It is a pleasure to let you know that your manuscript is now accepted for publication in Life Science Alliance. Congratulations on this interesting work.

DISTRIBUTION OF MATERIALS:

Again, congratulations on a very nice paper. I hope you found the review process to be constructive and are pleased with how the manuscript was handled editorially. We look forward to future exciting submissions from your lab.

Sincerely,
